# Influence of Arctic Stratospheric Ozone on Surface Climate in CCMI models

Ohad Harari[1], Chaim I Garfinkel[1], Shlomi Ziskin Ziv[1], Olaf Morgenstern[2], Guang Zeng[2], Simone Tilmes[3], Douglas Kinnison[3], Makoto Deushi[4], Patrick Jöckel[5], Andrea Pozzer[5], Fiona M. O'Connor[6], and Sean Davis[7]

[1]The Fredy and Nadine Herrmann Institute of Earth Sciences, Hebrew University of Jerusalem, Jerusalem, Israel.
[2] National Institute of Water and Atmospheric Research, Wellington, New Zealand.
[3] National Center for Atmospheric Research, Boulder, Colorado, USA
[4] Meteorological Research Institute, Tsukuba, Japan.
[5] Deutsches Zentrum für Luft- und Raumfahrt (DLR), Institut für Physik der Atmosphäre, Oberpfaffenhofen, Germany.
[6] Met Office Hadley Centre, Exeter, UK.
[7]NOAA Earth System Research Laboratory, Boulder, CO, USA.

*Correspondence to:* Chaim I. Garfinkel (chaim.garfinkel@mail.huji.ac.il)

**Abstract.** The Northern Hemisphere and tropical circulation response to interannual variability in Arctic stratospheric ozone is analyzed in a set of the latest model simulations archived for the Chemistry-Climate Model Initiative (CCMI) project. All models simulate a connection between ozone variability and temperature/geopotential height in the lower stratosphere similar to that observed. A connection between Arctic ozone variability and polar cap surface air pressure is also found, but additional statistical analysis suggests that it is mediated by the dynamical variability that typically drives the anomalous ozone concentrations. While the CCMI models also show a connection between Arctic stratospheric ozone and the El Niño Southern Oscillation (ENSO), with Arctic stratospheric ozone variability leading to ENSO variability one to two years later, this relationship in the models is much weaker than observed and is likely related to ENSO autocorrelation rather than any forced response to ozone. Overall, Arctic stratospheric ozone is related to lower stratospheric variability. Arctic stratospheric ozone may also influence the surface in both polar and tropical latitudes, though ozone is likely not the proximate cause of these impacts and these impacts can be masked by internal variability if data is only available for $\sim 40$ years.

## 1 Introduction

The stratospheric ozone layer not only protects life on Earth from solar ultraviolet radiation but also controls stratospheric temperature, which in turn affects tropospheric weather and climate (World Meteorological Organization, 2014). Stratospheric ozone was increasingly depleted in the last few decades of the twentieth century over many regions of the globe and is not expected to fully recover for many decades (World Meteorological Organization, 2014). Arctic stratospheric ozone (ASO) has been relatively spared from the worst ozone destruction due to the relatively stronger wave forcing from the troposphere leading to relatively warm temperatures as compared to the Antarctic, though during some winters, depletion has been observed (Staehelin et al., 2001) comparable to that in the Antarctic (e.g., March of 2011, Manney et al., 2011). It is well established

that ozone depletion in the Antarctic stratosphere affects surface climate (Polvani et al., 2011; Waugh et al., 2015), though a clear link between Arctic ozone depletion and surface climate has not yet been conclusively established.

Recent modeling studies have examined the possible connection between Arctic spring ozone and surface climate using a range of approaches, and obtained mixed results. Cheung et al. (2014) investigated whether the extreme Arctic ozone depletion of 2011 had an effect on tropospheric climate with the UK Met Office operational weather forecasting model, and found similar spring tropospheric forecast skill when forcing the model with more realistic ozone concentrations or with climatological ozone. Karpechko et al. (2014) found a connection between the 2011 low Arctic stratospheric ozone anomalies with tropospheric climate in atmospheric GCM simulations, but the connection was only robust if ozone anomalies are imposed together with sea surface temperature anomalies. They argue that the stratospheric response to reduced ozone is too weak if the anomalous tropospheric wave driving that initially led to the strong vortex is not also included, such that the radiative perturbation due to ozone requires feedbacks in order to robustly modulate surface climate (e.g., Kirchner and Peters, 2003). Smith and Polvani (2014) found that only for a prescribed ozone forcing larger than that historically observed is the tropospheric response robust in their simulations. In contrast, the coupled chemistry-climate simulations of Calvo et al. (2015) include a robust stratospheric-tropospheric response in low versus high ozone years if ozone depleting substance concentrations (ODS) follow those from 1985-2005: a positive phase of the North Atlantic Oscillation, a poleward shift of the North Atlantic tropospheric jet, and corresponding regional surface temperature anomalies. Downward coupling is not evident in the period 1955-1975 when ODS concentrations were lower, and they argue that the enhanced ODS concentrations led to enhanced dynamical variability of the vortex and thus to stronger downward coupling. The fully-coupled approach of Calvo et al. (2015) allows consistency between the evolving ozone distributions and dynamical conditions among other differences in the model configuration, which may explain the differences between their conclusions and those of studies prescribing ozone concentrations. However, there is some ambiguity in the approach of Calvo et al. (2015) as to whether the surface anomalies are due exclusively to chemical depletion of ozone: ozone anomalies are usually accompanied by anomalies in the Arctic vortex (i.e., early or delayed breakup of the Arctic vortex for high ozone or low ozone respectively, Hurwitz et al., 2011) and vortex anomalies independent of ozone have been shown to influence surface conditions (Black and McDaniel, 2007; Ayarzagüena and Serrano, 2009; Hardiman et al., 2011). Hence the degree to which the surface anomalies found by Calvo et al. (2015) are related to chemical ozone depletion rather than the altered dynamical state of the vortex which also affects ozone concentrations is ambiguous.

Observational studies have also suggested that interannual variability in ozone affects surface climate. Ivy et al. (2017) find that extreme Arctic stratospheric ozone (ASO) anomalies in March are associated with Northern Hemisphere tropospheric climate in spring (March-April) in specific regions of the Northern Hemisphere, with the effects generally consistent with the modeling study of Calvo et al. (2015). However, a delayed or advanced final warming of the Arctic vortex can lead to some of the surface impacts found by Ivy et al. (2017) (e.g., Black and McDaniel, 2007; Ayarzagüena and Serrano, 2009; Hardiman et al., 2011), and it is not clear whether the surface response is due to the dynamical impact from the final warming as opposed to the radiative impact of the ozone anomaly that typically accompanies a final warming. Finally, Xie et al. (2016) suggest that ASO anomalies influence sea level pressure anomalies over the North Pacific which in turn modulates subtropical

sea surface temperatures. This subtropical sea surface temperature anomaly might then lead to improved predictability of the El Niño Southern Oscillation, though this response is delayed by a year for reasons not yet clear (Garfinkel, 2017).

Overall, the very recent 2018 WMO Ozone Assessment Executive Summary appendix states that "there are indications that occurrences of extremely low springtime ozone amounts in the Arctic may have short-term effects on Northern Hemisphere regional surface climate" (World Meteorological Organization, 2018), and the goal of this paper is to explore the robustness of these indications. Specifically, we revisit the connection between boreal spring Arctic stratospheric ozone variability on inter-annual timescales and surface climate using the coupled ocean-atmosphere-chemistry models participating in the Chemistry-Climate Model Initiative (CCMI) project. We analyze $\sim 1722$ model-years of output data, which helps provide context for the associations evident in the relatively short observational record and in previous modeling studies which analyzed shorter simulations. The CCMI project is the first multi-model project in which many atmosphere-chemistry models were coupled to an interactive ocean, and hence the downward coupling in oceanic regions can be explored for the first time in a multi-model framework. We demonstrate that while Arctic ozone does appear to be associated with surface variability, the connection is largely associated with the dynamical control of ozone in the lower stratosphere by the polar vortex. In addition, while Arctic ozone does appear to influence ENSO for up to two years later, this association is much weaker than that observed and likely is associated with ENSO autocorrelation. Overall, while Arctic stratospheric ozone may lead to surface impacts, these impacts are generally weak and can be masked by internal variability if data is only available for $\sim 40$ years.

## 2    Data and Methods

### 2.1    Data

We analyze the Modern-Era Retrospective analysis for Research and Applications reanalysis (MERRA Rienecker et al., 2011), the merged ozone product from SWOOSH v2.6 from 1984 through 2014 (Davis et al., 2016), and output from atmospheric chemistry-climate-ocean general circulation models (CCMs) participating in the CCMI project.Full details of the satellite observations underlying SWOOSH and the interpolation scheme used can be found in Davis et al. (2016); here we use the combinedeqfillanomfillo3q product at 2.5 degree resolution with 31 vertical levels.

CCMI was jointly launched by the Stratosphere-troposphere Processes And their Role in Climate (SPARC) and the International Global Atmospheric Chemistry (IGAC) to better understand chemistry-climate interactions in the recent past and future climate (Eyring et al., 2013; Morgenstern et al., 2017). This modeling effort is an extension of CCMVal2 (SPARC-CCMVal, 2010), but utilizes up-to-date CCMs that also include tropospheric chemistry. The CCMI models used in this study are listed in Table 1. We consider the Ref-C2 simulations from models that are coupled to an interactive ocean and have uploaded their data to the British Antarctic Data Center server, and we also include the National Center for Atmospheric Research (NCAR) models (Morgenstern et al., 2017). Full details of the Ref-C2 simulations are described in Eyring et al. (2013); briefly, these simulations span the period 1960–2100, impose ozone depleting substances as in World Meteorological Organization (2011), and impose greenhouse gases other than ozone depleting substances as in RCP 6.0 Meinshausen et al. (2011). As we are interested in the surface impact of ozone anomalies over both land and ocean areas, we consider coupled ocean-atmosphere models

Table 1: Data products used

| data source | ensemble members | reference |
|---|---|---|
| MERRA/SWOOSH | 1 | Rienecker et al. (2011); Davis et al. (2016) |
| NIWA | 5 | Morgenstern et al. (2009) |
| CESM1 WACCM | 3 | Garcia et al. (2017) |
| CESM1 CAM4-chem | 3 | Tilmes et al. (2016) |
| HadGem3-ES | 1 | Hardiman et al. (2017) |
| MRI-ESM1r1 | 1 | Yukimoto et al. (2012) |
| EMAC | 1 | Jöckel et al. (2016) |

**Table 1.** The data sources used in this study.

only where surface impacts can occur in a self-consistent manner with the stratospheric ozone variability. In contrast, nearly all of the CCMVal2 models imposed sea surface temperatures (Morgenstern et al., 2010). The standard deviation of surface temperature variability in the Nino3.4 region for the models is shown in Figure 1, and the amount of variability in all models considered here is within 50% of that observed. One additional CCMI model included a coupled ocean (CHASER), however the ENSO in this model is too weak: the standard deviation of the Nino3.4 index in this model is 0.2 Kelvin as compared to the observed value of 0.85 Kelvin, and we therefore exclude it from this paper. Finally, we have examined the power spectral density for surface temperature in the Nino3.4 region in these models, and in all cases there is a peak between 2 and 5 years in general agreement with observations (not shown).

## 2.2 Methods

This study focuses on the impact of ASO on the troposphere on interannual timescales, and in order to remove any impacts on longer timescales due to climate change or ozone depleting substances, we first use multiple linear regression to remove the linear variability associated with greenhouse gases and ozone depleting substances from all time series (i.e., the same regression is applied to ozone, surface pressure, temperature, and polar cap heights). We use the equivalent $CO_2$ from the RCP6.0 scenario to track greenhouse gas concentrations (Meinshausen et al., 2011) and the effective equivalent stratospheric chlorine (EESC) following Newman et al. (2007, Eq.1, assuming a 5.5yr age-spectrum) to track ozone depleting substances on multi-decadal timescales. For consistency, this same multiple linear regression procedure is applied to MERRA/SWOOSH data.

Most models used in this study archive data over a period of 140 years, while the observational record available in either MERRA or SWOOSH extends for less than 40 years. To allow for a more natural comparison of model output to observations

we first perform the multiple linear regression as described above, and then divide the output from the CCMI models into periods of 41 years, to more closely match the data availability of MERRA/SWOOSH. Specifically, we divide the data of each of the CCMI models into three time periods: 1970-2010, 2011-2051, 2052-2092. The net effect is that we have 42 model-chunks of identical size (41 years) that we can meaningfully compare to observations (3 periods and 14 models). For each data source

we analyze the variables: temperature, geopotential height, surface air pressure, and the zonal mean ozone volume mixing ratio. We define a zonal mean index of Arctic stratospheric ozone (ASO) following Xie et al. (2016) as the area-weighted average of zonal mean ozone from 60°N to 81.25°N and mass-weighted from 150hPa to 50hPa. The poleward limit of the region used to define ASO is set at 81.25°N to match the data available from SWOOSH. For ENSO we use surface air temperature in the region bounded by 5°S-5°N and 190°E-240°E (i.e., the Nino3.4 region), as sea surface temperature was not available for all

models at the time we downloaded the data. Note that the observed negative correlation between ENSO and ASO at a lag of 20 months is essentially unchanged if we use air temperature as opposed to sea surface temperature.

The statistical significance of the correlation between two auto-correlated time series is computed using the two-tailed Student's t-test at the 95% confidence level: The effective number of degrees of freedom $N^{eff}$ used in the Student's-t test is approximated following Pyper and Peterman (1998), Li et al. (2013), and Li et al. (2012):

$$N^{eff} = \frac{1}{N} + \frac{2}{N} \times \sum_{j=1}^{N/4} \frac{N-j}{N} \times \rho_{xx}(j) \times \rho_{yy}(j) \qquad (1)$$

where N is the sample size, and $\rho_{xx}$ and $\rho_{yy}$ are the auto-correlation vectors of ENSO and ASO. The summation in equation 1 is performed up to $N/4$ following the recommendations of Pyper and Peterman (1998). In addition, Pyper and Peterman (1998) find that substituting $N^{eff}$ with $N^{eff} - 2$ provided a better balance between estimates of $N^{eff}$ and error rates, thus improving the validity of the student's t-test.

In order to assess the causality of the surface impacts contemporaneous with and following ASO anomalies, we form a causal effect network following Pearl causality (Pearl, 2000). The relative benefits and drawbacks of Pearl causality and Granger causality are discussed in Runge et al. (2017) and Samarasinghe et al. (2018). For our particular application we are interested in evaluating whether ASO or dynamical variability (which we track using area-weighted geopotential height from 80N to the pole at 100hPa, i.e., Zpole) leads to ENSO variability with leads of 10 to 27 months, i.e., whether ENSO has parents.

The causal effect networks analysis is based on a two-step algorithm. The first step is the PC algorithm (named after its developers, Peter Spirtes and Clark Glymour; Spirtes and Glymour, 1991), which is used to find the "parents" of ENSO from 10 to 27 months prior, with a null hypothesis that ENSO has no parents. The significance threshold used is $\alpha = 0.05$ (not the same $\alpha$ used in the t test - Runge et al. (2017)) and maximum combinations of conditions (i.e., $q_{max}$) is set to 10. The PC step starts by initializing the preliminary parents $P(X_t^j) = (X_{t-1}, X_{t-2}...X_{t-\tau_{max}})$ (where X includes the timeseries of ENSO,

ASO, and Zpole) and iteratively removes variables $X_{t-\tau}^i$ that do not pass the accepted significance threshold $\alpha$. It is tested again for different combinations of parents until the algorithm converges for a link $X_{t-\tau} \rightarrow X_t^j$ and the null hypothesis is rejected.

The second step of the algorithm is to evaluate the parents' causality strength, which is done with two different methods in order to assess robustness: MCI and Linear Mediation (Runge et al., 2015, 2017). The MCI step is based on partial correlations, and specifically calculates the correlation between ENSO and each of its parents after regressing out the influence of all other parents identified from the PC step. The statistical significance of the partial correlation result is tested with a two-tailed t test with $\alpha = 0.05$ (Runge et al., 2017). The Linear Mediation procedure as implemented here forms a multiple linear regression for ENSO using the parents identified in the PC step as regressors (Runge et al., 2015). In this step there is no statistical threshold. Thus, we get a score for all the parenting candidates and the coefficients of the multiple-linear regression procedure indicate the relative importance of each parent.

Full details of the algorithm are described in Runge et al. (2017) and a Python implementation of the algorithm as used in this study is the freely downloadable TIGRAMITE version 3.0 (https://jakobrunge.github.io/tigramite/). From the above-mentioned Python package we use two main functions: PCMCI and LinearMediation, the latter of which enables us to test sensitivity of the PCMCI results.

## 3  Effect of Arctic stratospheric ozone (ASO) on polar surface climate

We first consider the connection between ASO and zonally averaged temperature and geopotential height (Figure 2ab) in MERRA and SWOOSH data. Higher values of ASO are associated with elevated geopotential height and warmer temperature over the polar lower and mid-stratosphere, and lower geopotential heights and colder temperatures in the tropical lower and mid-stratosphere in reanalysis data (Figures 2ab). This effect is consistent with previous work that has shown that transport controls lower stratospheric ozone concentrations (Douglass et al., 1985; Hartmann, 1981; Rood and Douglass, 1985; Hartmann and Garcia, 1979; Silverman et al., 2017).

The CCMI ensemble-mean (i.e., the mean of all 42 sub-models) correlation between March ASO and March geopotential height and temperature is shown in Figure 3. The CCMI models capture the connection between the two phenomena, and the magnitude is similar to that observed though somewhat weaker in the lower stratosphere. In order to more meaningfully compare the relatively short observational record to the model output, we divide the model output into 41 year chunks (see section 2), and focus on coupling of ASO with polar cap 100hPa geopotential height and temperature in Figure 4a. The x-axis of Figure 4a shows the correlation of ASO with 100hPa polar cap temperature. The reanalysis is indicated with a yellow asterisk, the CCMI multi-model mean with a large black x, and each 41-year chunk of each model is indicated with a single x. In March, the multi-model mean correlation is weaker than that observed, but individual models simulate a tighter coupling than that observed. Strikingly, a different 41 year subsample of the same model can alternately simulate essentially no coupling of ASO with lower stratospheric temperatures (the 'x'es near the bottom left of the distribution) or coupling that is similar in magnitude to that observed. The y-axis of Figure 4a shows the correlation of ASO with 100hPa geopotential height in March. Models with a tighter connection between ASO and polar cap temperatures also feature a tighter connection between ASO and polar cap geopotential height, and the connection in reanalysis data falls well within that simulated by CCMI models. Results are similar for April as well (Figure 4b). This validates the fidelity of the coupling between these phenomena in the

CCMI models. It is beyond the scope of this paper (which focuses on monthly mean data) to determine the dominant direction of causality, though process-driven studies of observational data have indicated that lower stratospheric ASO anomalies are driven by transport(e.g., Douglass et al., 1985). Anomalous transport can occur both on large spatial scales where the same wind field that advects the ozone across the vortex barrier into the pole also leads to a warmer pole, and also on smaller scales where the causality of the connection between zonal mean temperature and ASO is less obvious. The strength of this connection is quantitatively similar for the period 1970-2010 when ODS concentrations were high and for the period 2052-2092 when imposed ODS concentrations are lower (Figure S43ab), suggesting that heterogeneous chemical ozone depletion plays a relatively minor role as compared to transport for the connection between ASO and polar vortex variability.

Does ASO affect the troposphere? Figure 6a shows the correlation of surface air pressure anomalies with ASO in March. Higher concentrations of ASO are associated with elevated surface air pressure anomalies over the polar cap and over Greenland, and with reduced surface air pressure further south in the Atlantic sector; overall the pattern resembles the negative phase of the North Atlantic Oscillation (consistent with Ivy et al., 2017, the feature over the North Pacific will be discussed in section 4). This relationship can be summarized by computing the correlation of surface air pressure anomalies area-weighted from 80N and poleward with ASO, and we show the result from MERRA and from all of the CCMI models on the x-axis of Figure 4c. The correlation in MERRA data is 0.41 (statistically significant at the 95% level), which is larger than in most, but not all, of the CCMI models. CCMI models show a stronger relationship in April instead (Figure 4d); the mean correlation across all models increases from 0.1 in March to 0.14 in April. While these correlations are statistically significant at the 95% level, the variance explained is low. Hence while the CCMI multi-model mean simulates a weaker connection between ASO and polar cap SLP in March than is observed, the observed association is enveloped by the range of CCMI models and the relationship between ASO and polar cap SLP in the CCMI models is still statistically significant at the 95% level (in agreement with Calvo et al., 2015).

What may explain the diversity in the strength of the coupling between ASO and polar cap SLP among the models? To provide a possible answer to this question, we repeat on the y-axis of Figure 4cd the correlation of ASO with polar cap height at 100hPa (the y-axis of the top row). There is a statistically significant relationship between coupling of ASO with 100hPa polar cap height and with the impact of ASO on the surface. Namely, 41-year model subsamples in which the connection between ASO and stratospheric polar cap height is stronger also simulate a stronger connection between ASO and polar cap SLP. Specifically, the correlation between these two is 0.43 in March and 0.58 in April (Figure 4cd), though clear outliers do exist. Hence the strength of the coupling between ASO and polar cap SLP is dependent on coupling between ASO and polar cap height at 100hPa. This dependence suggests that there is ambiguity as to whether ASO is indeed the proximate cause for the surface climate anomalies evident on the x-axis of Figure 4cd. Specifically, ozone anomalies are usually accompanied by anomalies in the Arctic vortex (Figure 4ab) and previous work has shown that spring Arctic vortex anomalies independent of ozone can influence surface conditions (Black and McDaniel, 2007; Ayarzagüena and Serrano, 2009; Hardiman et al., 2011). We now demonstrate that vortex anomalies are associated with polar cap SLP anomalies in the CCMI models as well, and then try to isolate statistically the relative importance of ASO.

We first show that stratospheric polar cap geopotential height and temperature are even more strongly related to tropospheric variability than is ASO. The y-axis of Figure 4ef considers the correlation of 100hPa polar cap height with polar cap SLP across the CCMI models as compared to that observed. The correlation of 100hPa polar cap geopotential height with polar cap SLP is more than double the corresponding correlation of polar cap SLP with ASO for both the multi-model mean of the CCMI models (0.57 as compared to 0.1 in March, and 0.39 as compared to 0.14 in April), and for reanalysis data. Results are similar for the correlation of 100hPa polar cap geopotential temperature with polar cap SLP (on the x-axis): the correlation of ASO with polar cap SLP is also weaker than that of 100hPa polar cap temperature with polar cap SLP. This effect is highlighted visually in Figure 5, which compares 100hPa polar cap height with polar cap SLP for each model integration on the top row and ASO with polar cap SLP on the bottom row. It is clear that the connection between 100hPa geopotential height and surface conditions is far stronger than that between ASO and surface conditions.

It could well be that the apparent connection between ASO and polar cap SLP is just a byproduct of the tight connection between polar cap stratospheric geopotential height with both parameters, rather than a direct connection. We address this issue by using linear regression to statistically remove the portion of polar cap SLP variability and ASO variability that is linearly related to stratospheric polar cap geopotential height for each model, and then consider whether there remains any lingering connection between ASO and polar cap SLP. The results are displayed on the x-axis of the bottom row of Figure 4. The correlation of ASO with polar cap SLP is now negative both for most CCMI models and also for observations - high ozone is associated with lower heights over the pole - opposite of the relationship when Z was not regressed out. This statistical argument indicates that there is no distinct pathway whereby ASO affects the polar troposphere, but rather it affects the polar troposphere through its coupling to the dynamics of the lower stratospheric polar vortex.

The bottom row of Figure 4 uses linear techniques to deduce that any influence of ASO on surface climate is mediated through the dynamics of the lower stratospheric polar vortex; however, there could be feedbacks between ASO, the lower stratospheric polar vortex, and surface climate. For example, the x-axis of Figure 7 shows the correlation between polar cap temperature and polar cap surface pressure for the period 1970-2010 in blue, for the period 2011-2051 in green, and for the period 2052-2092 in red. The correlation between polar cap temperature and surface conditions is stronger in the future when ODS concentrations are minimal and heterogeneous chemical ozone depletion less common (among other changes in the climate). Similar results are found for the correlation of polar cap geopotential height and polar cap surface pressure (y axis), and also for the upper troposphere (not shown). Indeed, the correlation of ASO with polar cap surface pressure is also marginally higher in a future with low ODS concentrations than in the historical period (Figure 5), further suggesting that heterogeneous chemical ozone depletion is not a crucial factor for a strong downward impact. We leave for future work an explanation for this relationship.

## 4 Effect of Arctic stratospheric ozone (ASO) on ENSO

Xie et al. (2016) recently suggested that Arctic stratospheric ozone anomalies influence North Pacific sea level pressure anomalies which in turn affects subtropical sea surface temperatures, and the subtropical sea surface temperature anomalies modulate

ENSO through the seasonal footprinting mechanism approximately 20 months later. Their conclusions were based on the limited observational record and model experiments with one model, and we now consider whether CCMI models capture this association in order to assess the robustness of this effect. The CCMI models included in our study contain both ozone variability and an internally generated ENSO (Figure 1), hence they are the first multi-model ensemble which simulates the relevant underlying processes.

We begin with the lagged-correlation of ASO and ENSO for lags ranging between -40 to 40 months using MERRA/SWOOSH data (figure 8a) following Xie et al. (2016) in order to establish context for the CCMI models. ENSO is positively correlated with ASO seven months later (at lag equal to -7), such that e.g., El Niño leads to more ozone seven months later. The more robust relationship is a negative correlation between ASO and ENSO 20 months later (at lag equal to 20) such that e.g., more ASO leads to a La Niña 20 months later, though this relationship is not statistically significant at the 95% level using a two-tailed Student's t-test. Xie et al. (2016) noted that the strongest observed connection between ASO and ENSO is March ASO with ENSO 20 months later, and hence we show the lagged-correlation between March ASO and ENSO in figure 8b. The correlation of ENSO in January with ASO two months later in March is 0.2, and the correlation of ASO in March with ENSO $\sim 18$ months later is nearly -0.5. All of these relationships are in agreement with Xie et al. (2016), though we find that this relationship is slightly less statistically significant than Xie et al. (2016).

Do the CCMI models capture these relationships between ASO and ENSO? We first focus on the multi-model mean lagged correlation between ENSO and ASO in Figure 9. The lagged-correlation function between ASO and ENSO reveals the same general lead/lag behavior as seen in the MERRA/SWOOSH model but generally with much weaker correlations. The positive correlation at lag=-4 (Figure 9a) exceeds 0.1, corresponding to El Niño leading to enhanced ASO after 4 months, and this effect is highly statistically significant and is consistent with that found in CCMVal models (Cagnazzo et al., 2009). Hence the models are able to capture the forcing of ASO anomalies by ENSO, with El Niño leading to enhanced ASO and La Niña leading to reduced ozone (Cagnazzo et al., 2009).

The lagged-correlation also contains a negative peak at a lag of +10 months, and statistical significance is maintained from a lag of 6 months up to a lag of 22 months. This result is in agreement with the feature seen in MERRA/SWOOSH and Xie et al. (2016). However the correlation at positive lags in the multi-model mean is a factor of five weaker than that observed; furthermore, the correlation is strongest not 20 months after the ASO anomalies, but rather 10 months later. We have also computed the correlation coefficients between March ASO and ENSO in order to focus on the season where the observed relationship peaks (Figure 9b). A statistically significant negative correlation is found when ASO lead ENSO by 8-14 months. An additional, smaller yet also statistically significant, negative local peak is obtained at a lag of 20 month (at Nov(1) on Figure 9b), qualitatively in agreement with the observed relationship but much weaker in magnitude.

The multi-model mean lagged-correlations differ from that observed in two aspects: they are weaker and also peak at earlier lags. This does not necessarily imply that the models are inconsistent with the observed effect, as some models do show a relationship that resembles that observed. To highlight this effect, we compare adjacent 41 year sub-samples for a given model. The results are shown in the bottom four rows of Figure 8. Figure 8cegi is constructed analogously to Figure 9, but it focuses on two different 41 year sub-samples from the same integration for two models. These two sub-samples indicate an opposite

lead/lag relationship between ENSO and ASO between adjacent sub-samples for the same integration. For example, WACCM run #3 over the years 2011-2051 indicates a similar lead-lag relationship as the one observed in MERRA/SWOOSH (figure 8c). Yet, over the years 2052-2092 (Figure 8e), the exact same model integration suggests that any apparent modulation of ENSO by ASO is weak and only appears at much shorter lags. Another example for the intra-model variability is evident in the results of the NIWA-R3 model (Figure 8gi). NIWA-R3 simulates a lead-lag relationship that does not resemble the observed one over the years 2011-2051, whereas for the years 2052-2092 the lead/lag relationship shows some similarities. Thus, these two sub-samples of the same model are different from each other and therefore gives different results when comparing to MERRA/SWOOSH. Similar results are evident if we focus on the lagged correlation of March ASO with ENSO (right column of Figure 8).

The model spread in the connection of ASO with ENSO 10-20 months later does not appear to be associated with stratospheric dynamical or temperature variability, as e.g., the correlation between ASO and stratospheric temperature and geopotential height is qualitatively similar in these experiments. Specifically, these specific sub-samples from WACCM (compare Figures 2cd to Figures 2ef) and NIWA (compare Figures 2gh to Figures 2ij) both show a positive correlation between ASO and polar cap geopotential and temperature. Hence internal tropospheric or oceanic variability appears to be the source of the difference in the connection between ASO and ENSO. That internal tropospheric or oceanic variability can mask the connection between ASO and ENSO over a 41 year period suggests that the observational polar ozone record is too short to reach firm conclusions as to the connection between ENSO and ASO. More specifically, it is possible that internal climate variability could have contributed to the apparent observed effect. The possible role of internal variability can be reduced as much as possible by computing the multi-model mean of the correlations (by averaging over all lagged-correlation coefficients), and as discussed above the multi-model mean correlation is much weaker and peaks sooner than that observed.

Xie et al. (2016) propose a specific mechanism between ASO and ENSO, and we now evaluate whether this mechanism is operating in the CCMI models. Specifically, they argue that higher ASO leads to higher SLP over the North Pacific in March and April (red box on Figure 6ab), which directly leads to warmer sea surface temperatures in the subtropical North Pacific (red box on Figure 6c) as the cold continental winds off Eurasia are weakened. This warming of the subtropical North Pacific then leads to a La Niña event due to the seasonal footprinting mechanism (Vimont et al., 2003). It is difficult to deduce any evidence for these effects in the multi-model mean surface pressure and sea surface temperature response (Figure 10), though the multi-model mean correlation between ASO and ENSO was also weak. Even if we focus on the model sub-samples that did succeed in capturing a relationship between ASO and ENSO (e.g., WACCM run #3 for the years 2011-2051 and NIWA-R3 over the years 2052-2092), there is no better agreement with the observed SLP and surface temperature anomalies than in the model sub-samples that failed to capture the observed relationship between ASO and ENSO (Figure 6). The correlation between April surface temperature in the red boxed region of Figure 6c with March ASO averaged across all CCMI models is 0.02, and while individual model sub-samples simulate correlations as large as the observed correlation of 0.3, there is no significant relationship between the models that simulate greater warming in this region in response to enhanced ASO and the relationship between ASO and ENSO 20 months later. Hence there is no evidence that the mechanism of Xie et al. (2016) is present in the CCMI models.

One might hypothesize that the apparent relationship between ASO and ENSO $\sim 20$ months later is driven by the auto-correlation of ENSO: El Niño (which typically drives higher ASO) is often followed by a La Niña event a year or two later, and one might therefore suppose that the La Niña event 20 months after the high values of ASO is due to internal oceanic ENSO dynamics and does not involve the stratosphere (e.g., Garfinkel, 2017).

We test this hypothesis using Pearl causality as described in Section 2.2; briefly, we test whether ENSO has a "parent" ASO. The results of the causality test are shown in Figure 11, a heat map of parents of ENSO where only statistically significant parents are indicated. Figure 11a shows results for the PCMCI algorithm, and Figure 11b shows results for the Linear mediation algorithm (see section 2.2).

For the observational data (i.e., SWOOSH 1984-2014) we can see that ENSO has two parents: ASO 20 and 22 month prior to ENSO. This connection is obtained from both MCI and Linear Mediation, and it is a negative connection. This result is in agreement with both Xie et al. (2016) and our correlation results (Figure 8a), albeit with lower regression coefficients and correlations: for the MCI step we get a score of -0.17 for ASO(-20) and -0.19 for ASO(-22), and in the linear mediation -0.21 for ASO(-20) and -0.22 for ASO(-22). These regression coefficients are lower than the simple lag-correlation of ENSO with ASO (-0.32 in our study and -0.35 in Xie et al. (2016), both for ASO (-20)).

In the CCMI ensemble, ASO is not a parent of ENSO over the historical period for any model for lags near -20 (and in fact is a parent with the opposite sign in one model). In contrast, ENSO(-10) is a parent for half of the models, and for these models both the MCI step and the linear mediation give similar results. For later time periods, ASO is a parent of ENSO for some models for lags near -20, such that the observed relationship between ASO(-20) and ENSO is simulated on occasion, though in other models ENSO(-10) is a more important parent. Hence over the period 2011 through 2092 the tendency noted by Xie et al. (2016) is evident in some of the CCMI models, though the magnitude of the connection is much weaker than observed.

The association between ASO and ENSO in the CCMI models does not appear to be related to dynamical changes in the Arctic vortex. The red lines in Figure 8b and Figure 9b indicate the lagged correlation between March 100hPa polar cap geopotential height and ENSO. El Niño is associated with higher March geopotential height at zero lag in the CCMI models (though not in observations over this time period; Hu et al., 2017; Domeisen et al., 2019; Garfinkel et al., in press), but there is no indication that higher geopotential height (typically associated with high values of ASO) leads to La Niña a year later. Furthermore, Zpole is not a parent of ENSO for any lead between 10 months and 27 months in observations, and is also not a parent for most models either. (Recall that polar cap SLP is more strongly affected by polar cap geopotential height than ASO.)

Overall, the CCMI models do show a tendency of ASO variability to lead ENSO variability, but the effect is much weaker than that observed. This does not imply that the models are inconsistent with observations, as individual models do capture associations as strong as that observed. Rather, internal climate variability could be contributing to the apparent observed connection between ASO and ENSO.

## 5  Conclusions

The effects of Antarctic stratospheric ozone depletion on surface climate are well studied and have been shown to play a dominant role in recent trends of Southern Hemisphere surface climate (Polvani et al., 2011; World Meteorological Organization, 2014; Waugh et al., 2015). While trends in Arctic stratospheric ozone (ASO) are weak, interannual variability in the Arctic is larger than in the Antarctic. However, the tropospheric impacts of this interannual variability in the Arctic stratospheric ozone layer have proven difficult to isolate, with different studies reaching opposite conclusions. Here we use the CCMI models to establish context for the observed connection between interannual ASO changes and polar tropospheric variability and ENSO. We focus on the CCMI models for two main reasons: First, $\sim 1722$ model-years of output data are available, which helps provide context for the associations evident in the relatively short observational record and in previous modeling studies which analyzed shorter simulations. Second, the CCMI project is the first multi-model project in which many atmospheric models were coupled to an interactive ocean and chemistry, and hence the downward coupling in oceanic regions can occur in a physically meaningful manner.

Increased Arctic stratospheric ozone is associated with an increase in both polar cap temperatures and geopotential height over the polar lower and mid-stratosphere, in the CCMI models. Models with a stronger connection between ASO and polar cap temperatures also tend to simulate a stronger connection between the ASO and polar cap geopotential height. The strength of the connection between ASO and polar stratospheric temperature and height in the CCMI models straddles the strength of the observed connection between ASO and polar stratospheric temperatures and heights, suggesting that there is no discrepancy between the models and the observations.

ASO was also found to be significantly correlated with polar cap surface pressure anomalies, in agreement with Calvo et al. (2015). However these correlations are weak in the multi-model mean, and hence ASO may not be particularly useful for prediction of surface climate. Furthermore, the proximate cause of this surface impact is ambiguous as polar cap height anomalies can also lead to surface impacts. In fact, the association between polar cap surface pressure anomalies and 100hPa polar cap geopotential height is stronger than that between polar cap surface pressure anomalies and ASO, and if we regress out from polar cap surface pressure anomalies any linear influence associated with 100hPa polar cap geopotential height, then there is no apparent relationship with ASO. This suggests that there is no distinct effect of ASO on the polar troposphere per se, and rather the connection between ASO and tropospheric variability is mediated through its covariability with the Arctic stratospheric vortex. The presence of realistic coupling between ozone and wind/temperature fields may account for the difference in conclusion between the study of Calvo et al. (2015) - who find a robust connection between ASO and the surface in a model which simulates ozone-dynamics interactions - and those of Cheung et al. (2014), Karpechko et al. (2014), and Smith and Polvani (2014) who find little tropospheric response to a fixed ozone perturbation decoupled from the dynamics in the stratosphere.

Enhanced ASO appears to lead to La Niña 10 to 20 months later in observations (in agreement with Xie et al. (2016)) and in the CCMI models, though the association in the CCMI models is much weaker than that observed, strongest at earlier lags, and likely associated with the autocorrelation of ENSO and not actually forced by ASO per se. Furthermore, the exact same CCMI model integration can alternately simulate an association quantitatively similar to that observed or no association.

Rather, internal climate variability can cloud any connection between ENSO and ASO when only 40 years of data are available, and much longer time-periods are needed to average over the large internal variability present. Finally, some of the observed connection of ASO with ENSO is due to autocorrelation of ENSO, as when we form a multiple linear regression to predict ENSO using ENSO at -10 months lag in addition to ASO at lags of near -20 months, the regression coefficient of ASO with ENSO drops by around one-third if ENSO autocorrelation is included.

This work raises several questions for future work. First, establishing the direction of causality between stratospheric polar cap ozone and temperature/height anomalies in the CCMI is beyond the scope of this work, as daily data is needed to resolve the key processes. However previous work has suggested that dynamical transport drives ozone anomalies in the lower stratosphere (Douglass et al. (1985), Hartmann (1981), Rood and Douglass (1985), Hartmann and Garcia (1979), Silverman et al. (2017)). Such transport occurs in large scale eddies and in polar downwelling that includes a balanced temperature perturbation, and also in smaller scale mixing across the vortex edge where the wind field and temperature field are not necessarily balanced. Second, Calvo et al. (2015) argue that chemical ozone depletion leads to a more variable vortex and specifically to more extremely strong vortex events, which in turn leads to more robust stratosphere-troposphere coupling. However we find that variability of the vortex is actually larger in the future when ODS concentrations are lower and heterogeneous chemical ozone depletion should occur less frequently. Specifically, the standard deviation averaged across all models for the period 1970-2010 is lower than for the period 2052-2092 for ASO, polar cap geopotential height, and polar cap height by 15% to 20%, and this increase in variability is driven by both more frequent positive and negative extremes (e.g. Figure 5). A thorough investigation of what drives this enhanced future stratospheric vortex variability in the CCMI models is beyond the scope of this work, though we note that Ayarzagüena et al. (2018) find little future change in sudden warming frequency in these models. Third, the robustness of the effect of ASO on ENSO is still unclear. The observed effect between them does appear robust when evaluated in the framework of Pearl causality, however the connection is not robust in the CCMI models once other potential parents of ENSO are taken into account. However, the causal effect network we implemented only assesses linear relationships, and hence any nonlinear impact may be missed. Fourth, it is conceivable that the range of variability in stratospheric dynamics (i.e., the NAM) may be larger if ozone is interactive than when it is not, as a similar effect appears to be present in the Southern Hemisphere (Dennison et al., 2015). Finally, the mechanism whereby polar stratospheric variability (whether dynamical or in ozone) influences the tropospheric circulation is beyond the scope of this work, though we suspect that it will be very difficult to tease out mechanisms from the CCMI models due to tropospheric feedbacks reinforcing any initial response forced by the stratosphere (Garfinkel et al., 2013; Garfinkel and Waugh, 2014; Kidston et al., 2015).

Overall, (1) there is a strong connection between ASO and lower stratospheric temperature and geopotential height, and this connection likely mediates the connection between ASO and tropospheric changes; (2) the CCMI models capture most aspects of the connections that have been found in observations though not necessarily with the same magnitude as that observed, with differences possibly due to internal variability; and (3) ASO may contain predictive information for ENSO up to two years later, but any additional skill ASO contributes is modest.

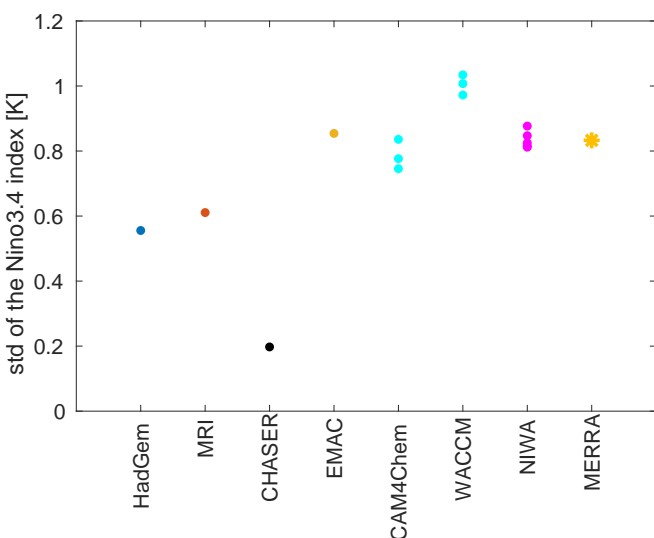

**Figure 1.** Standard deviation of surface temperature in the Nino3.4 region in the CCMI models. The multiple dots for NIWA, WACCM and CAM4Chem represent different integrations of the same model. The standard deviations in CHASER is much less than that observed, and hence this model was excluded from further analysis.

*Author contributions.* OH performed all data analysis and helped prepare the manuscript. CIG helped prepare the manuscript and provided overarching guidance and ideas. SZZ performed the Pearl Causality calculation. The other co-authors carried out the numerical experiments and provided feedback on the paper.

## 6   Acknowledgment

CIG was supported by the Israel Science Foundation (grant number 1558/14) and by a European Research Council starting grant under the European Union's Horizon 2020 research and innovation programme (grant agreement No 677756). We thank the international modelling groups for making their simulations available for this analysis, the joint WCRP SPARC/IGAC CCMI for organizing and coordinating the model data analysis activity, and the British Atmospheric Data Centre (BADC) for collecting and archiving the CCMI model output. All datasets used in this study are available online: |http://blogs.reading.ac.uk/ccmi/badc-data-access|. We thank the two anonymous reviewers

for their helpful comments, and Marie Mcgraw and Marlene Kretschmer for informative discussions on Granger and Pearl causality. The TIGRAMITE package used to calculate causality can be downloaded from |https://jakobrunge.github.io/tigramite/|. Correspondence and requests for data should be addressed to C.I.G. (email: chaim.garfinkel@mail.huji.ac.il).

F.M. O'Connor and the development of HadGEM3-ES was supported by the joint DECC/Defra Met Office Hadley Centre Climate Programme (GA01101).

The EMAC simulations have been performed at the German Climate Computing Centre (DKRZ) through support from the Bundesministerium für Bildung und Forschung (BMBF). DKRZ and its scientific steering committee are gratefully acknowledged for providing the HPC and data archiving resources for the consortial project ESCiMo (Earth System Chemistry integrated Modelling).

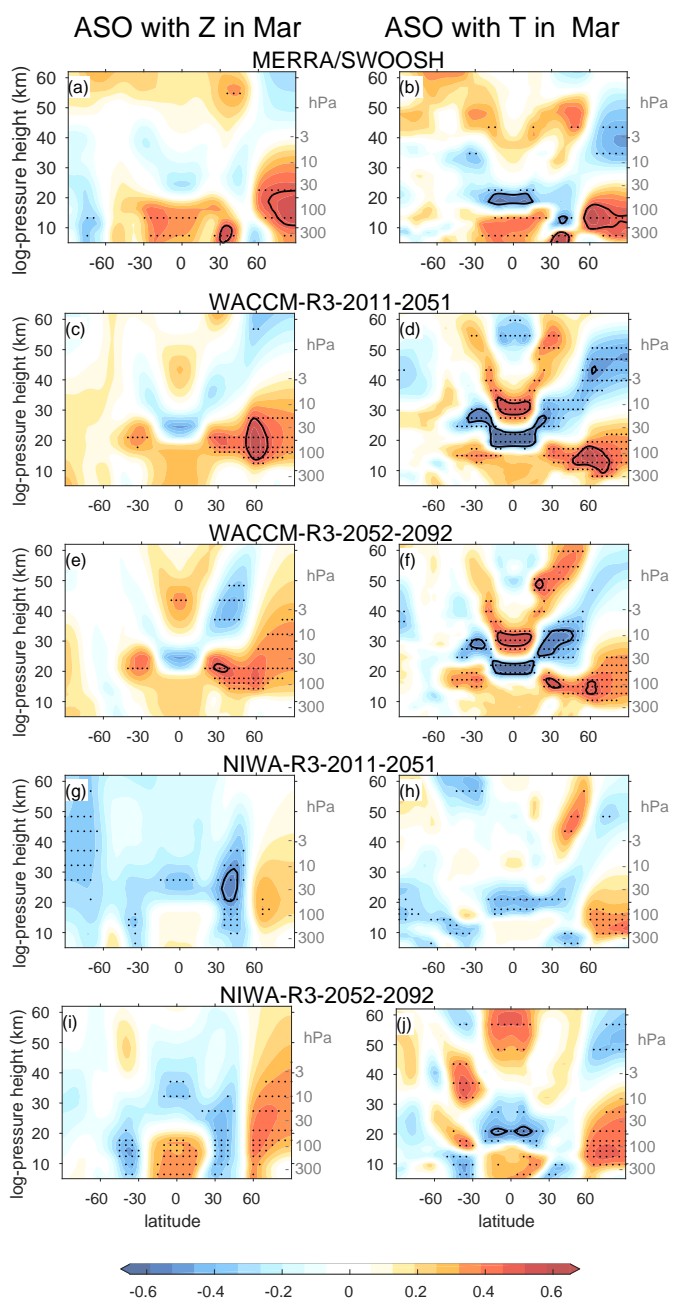

**Figure 2.** Correlation coefficients between ASO and geopotential height anomalies (left column) and between ASO and temperature anomalies (right column) for each of the examined data sources: (a-b) MERRA/SWOOSH, (c-f) WACCM-R3, (g-j) NIWA-R3. The black dots represent locations where correlations are statistically significant at the 95% level (see section 2.2). Correlations of ±0.5 are thin black. The contour interval is 0.065

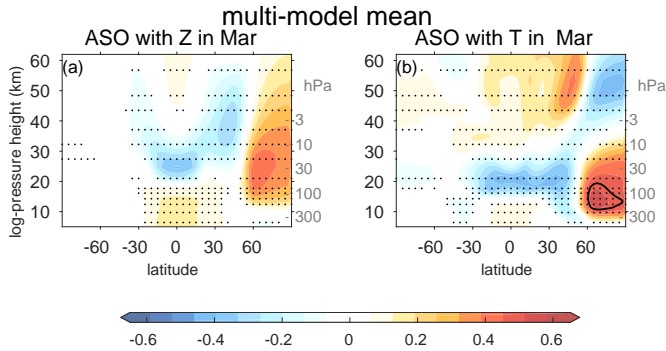

**Figure 3.** As in Figure 2 but for the multi-model mean

For the WACCM results, computing resources (ark:/85065/d7wd3xhc) were provided by the Climate Simulation Laboratory at NCAR's Computational and Information Systems Laboratory, sponsored by the National Science Foundation and other agencies. OM and GZ acknowledge the UK Met Office for use of the Unified Model, the NZ Government's Strategic Science Investment Fund (SSIF), and the contribution of NeSI high- performance computing facilities to the results of this research (https://www.nesi.org.nz). OM also acknowledges funding by the New Zealand Royal Society Marsden Fund (grant 12-NIW-006).

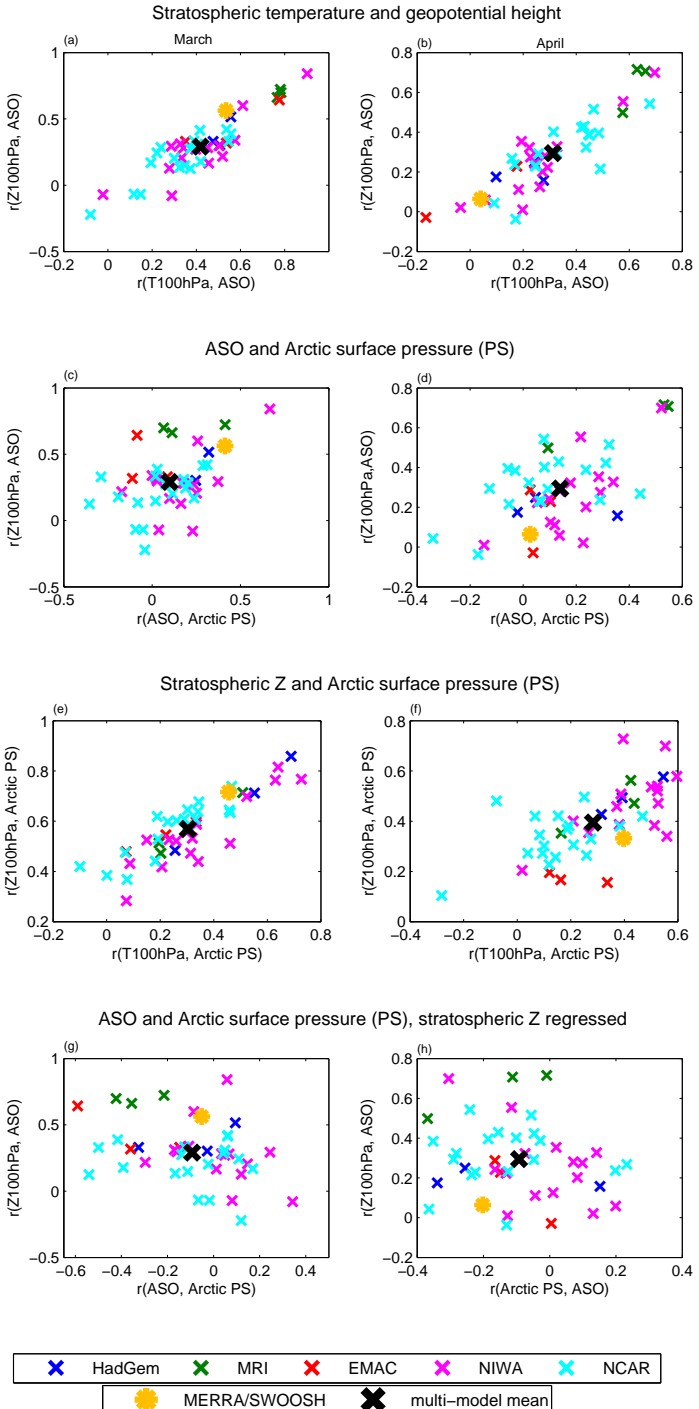

**Figure 4.** The connection between correlations of ASO anomalies and anomalies in various metrics of the stratosphere and troposphere. In each subplot the X- and Y-axis represent correlation of two measures. Each x represents a different 41 year chunk of model output, and reanalysis/observational data is represented by asterisks. The first row compares the correlation of ASO with polar cap stratospheric temperature(T) with the correlation of ASO with polar cap geopotential height(Z) in (a) March and in (b) April. The second row compares the correlations of ASO and polar cap height (Z) with ASO and polar cap surface pressure(PS). The third row compares the correlations of polar cap Z and PS with polar cap T with PS. The forth and final row is similar to the second row except that polar cap Z is regressed out.

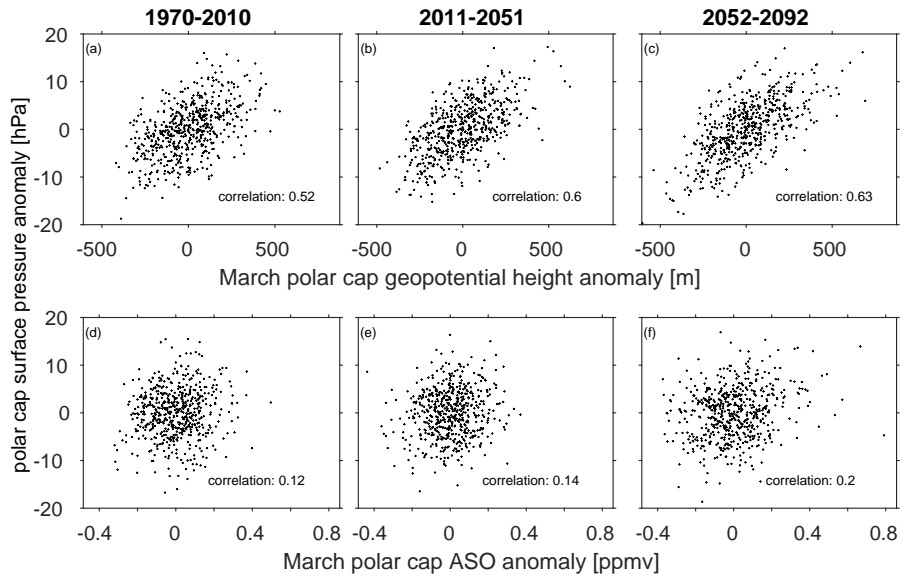

**Figure 5.** A scatter plot of (y-axis) polar cap surface pressure with March (x-axis; top) polar cap geopotential height and (x-axis; bottom) Arctic stratospheric ozone. April polar cap surface pressure is used for the top row, and March polar cap surface pressure is used for the bottom row, corresponding to the month with strongest coupling. Each year of each CCMI integration is marked with a diamond. All correlations are statistically significant at the 95% level as given by a two-tailed student-t test, as are the differences in correlation between the top row and bottom row.

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

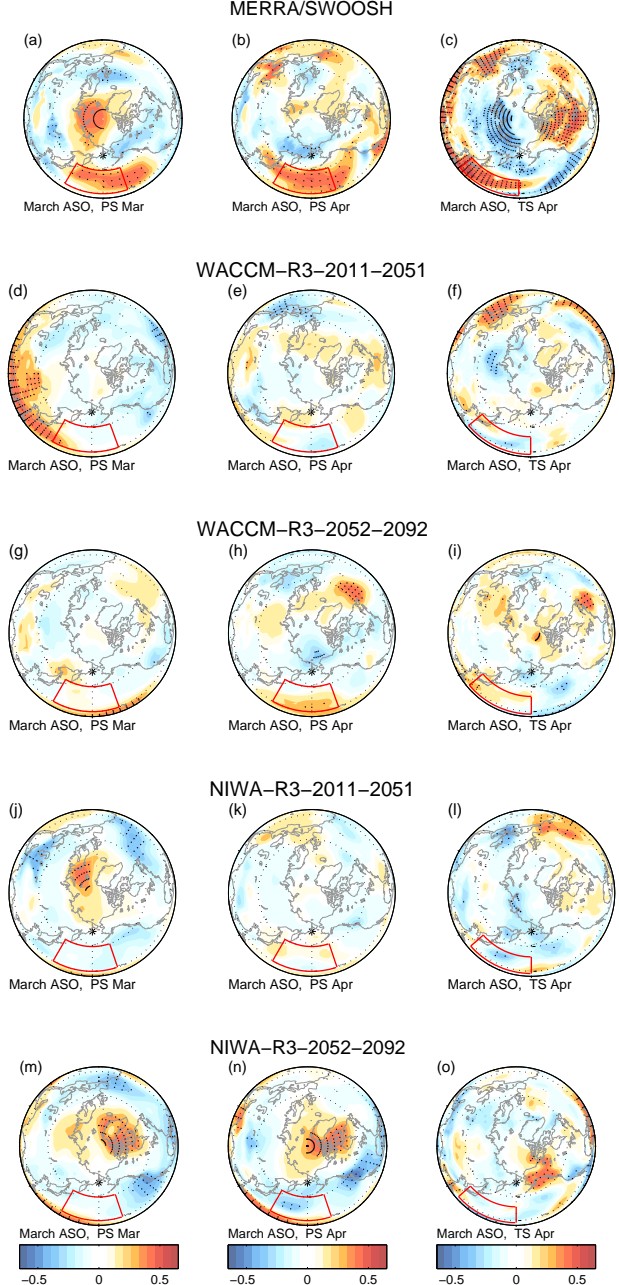

**Figure 6.** Correlation coefficient between ASO anomalies and surface pressure (PS) or surface temperature (TS) anomalies, for MERRA/SWOOSH, WACCM-R3, and NIWA-R3, where R3 indicates the third realization. The left and center columns represent the correlation between ASO and surface pressure. The region of interest discussed in section 4 is marked by the red frame, and is defined by 20-50N and 150-200W. The leftmost column shows the correlation between March ASO and March surface pressure, and the center column shows the correlation between March ASO and April surface pressure. The rightmost column shows the correlation between the March ASO anomalies and April near-surface temperature. The region of interest discussed in section 4 is marked by a red frame, and is defined by 15-40N and 130-180W. In all the plots, the black dots mark the areas in which the correlation is statistically significant at the 95% level. The contour interval is 0.065.

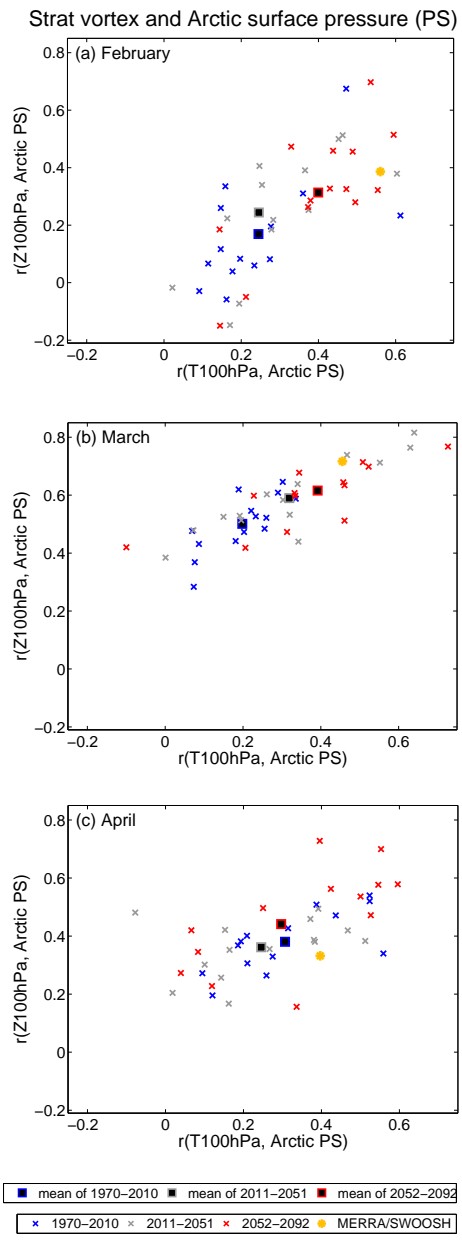

**Figure 7.** Correlation between stratospheric variability and polar cap surface pressure(PS), with simulations conducted during the period 1970-2010, 2011-2051, 2052-2092 in separate colors in (a) February , (b) March, and (c) April. For the x-axis, the correlation is conducted with polar cap temperature at 100hPa, and for the y-axis, the correlation is conducted with polar cap geopotential height at 100hPa.

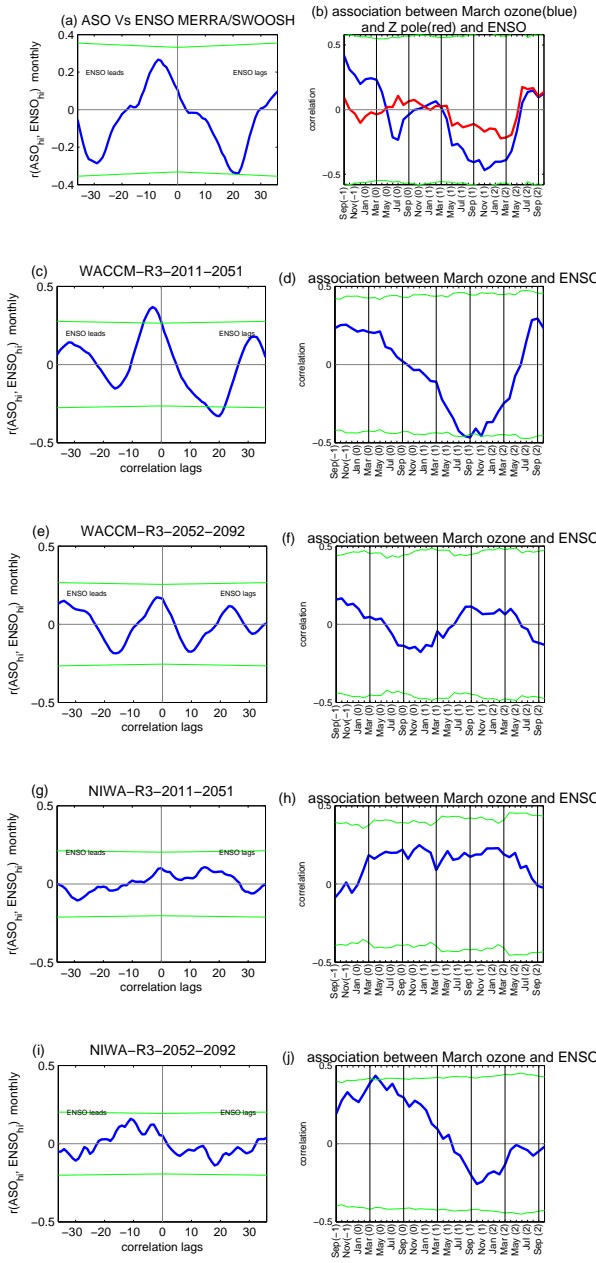

**Figure 8.** Lagged-correlation between ASO and ENSO for (a-b) MERRA/SWOOSH, (c-f) WACCM-R3, (g-j) NIWA-R3. The left column shows the lagged-correlation for all calendar months, and the right column shows the association between March ASO of every year and ENSO of every month across all years. The green lines mark the student's t-test 95% confidence level - correlations that exceed the upper green line or are more negative than the lower green line are statistically significant. The red line in subplot (b) represents the correlation between March polar geopotential height and ENSO.

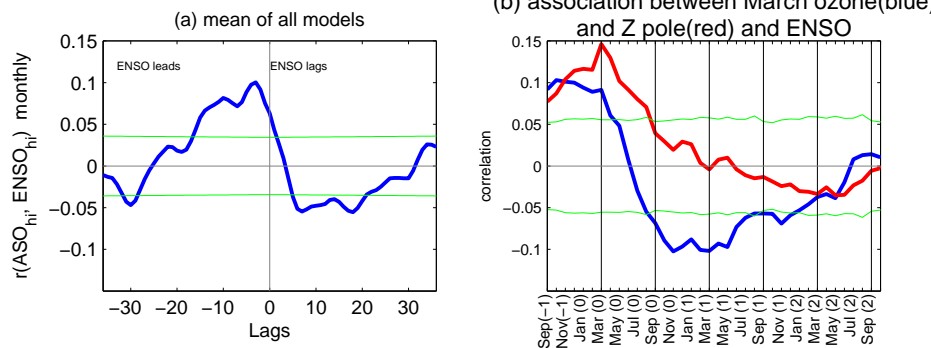

**Figure 9.** As in Figure 8 but for the multi-model mean.

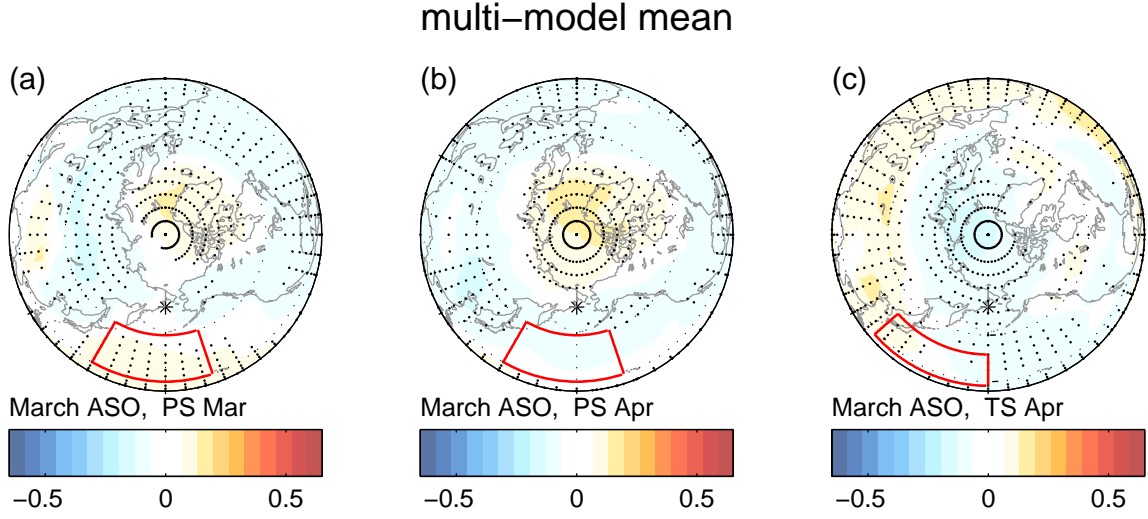

**Figure 10.** As in Figure 6 but for the Multi-model Mean.

Cheung, J., Haigh, J., and Jackson, D.: Impact of EOS MLS ozone data on medium-extended range ensemble weather forecasts, Journal of Geophysical Research: Atmospheres, 119, 9253–9266, 2014.

Davis, S. M., Rosenlof, K. H., Hassler, B., Hurst, D. F., Read, W. G., Vömel, H., Selkirk, H., Fujiwara, M., and Damadeo, R.: The Stratospheric Water and Ozone Satellite Homogenized (SWOOSH) database: A long-term database for climate studies, Earth System Science Data, 8, 461, 2016.

Dennison, F. W., McDonald, A. J., and Morgenstern, O.: The effect of ozone depletion on the Southern Annular Mode and stratosphere-troposphere coupling, Journal of Geophysical Research: Atmospheres, 120, 6305–6312, doi:10.1002/2014JD023009, https://agupubs.onlinelibrary.wiley.com/doi/abs/10.1002/2014JD023009, 2015.

Domeisen, D. I., Garfinkel, C. I., and Butler, A. H.: The Teleconnection of El Niño Southern Oscillation to the Stratosphere, Reviews of Geophysics, 2019.

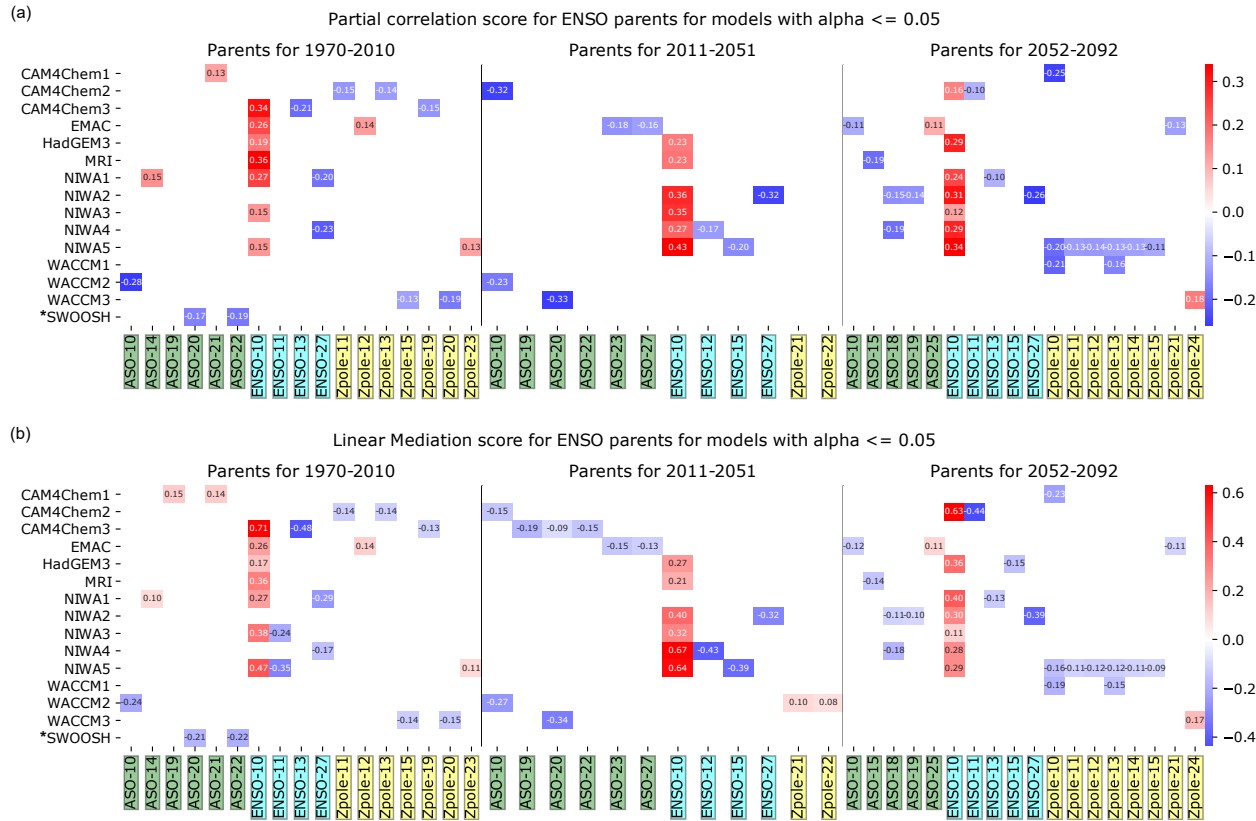

**Figure 11.** Pearl Causality analysis of the parents of ENSO using the (a) PCMCI algorithm and (b) using the PC algorithm with linear mediation. See Section 2.2 for details of the algorithm. Observations are in the bottom row, and each of the CCMI models is shown separately. Parents of ENSO are shown in color, with darker shades for more robust parents.

Douglass, A. R., Rood, R. B., and Stolarski, R. S.: Interpretation of ozone temperature correlations: 2. Analysis of SBUV ozone data, Journal of Geophysical Research: Atmospheres, 90, 10 693–10 708, 1985.

Eyring, V., Arblaster, J., Cionni, I., Sedláček, J., Perlwitz, J., Young, P., Bekki, S., Bergmann, D., Cameron-Smith, P., Collins, W. J., et al.: Long-term ozone changes and associated climate impacts in CMIP5 simulations, Journal of Geophysical Research: Atmospheres, 118, 5029–5060, doi:10.1002/jgrd.50316, 2013.

Garcia, R. R., Smith, A. K., Kinnison, D. E., Cámara, Á. d. l., and Murphy, D. J.: Modification of the Gravity Wave Parameterization in the Whole Atmosphere Community Climate Model: Motivation and Results, Journal of the Atmospheric Sciences, 74, 275–291, 2017.

Garfinkel, C. I.: Might stratospheric variability lead to improved predictability of ENSO events?, Environmental Research Letters, 12, 031 001, 2017.

Garfinkel, C. I. and Waugh, D. W.: Tropospheric Rossby Wave Breaking and Variability of the Latitude of the Eddy-driven Jets, Journal of Climate, 27, doi:10.1175/JCLI-D-14-00081.1, 2014.

Garfinkel, C. I., Waugh, D. W., and Gerber, E. P.: The Effect of Tropospheric Jet Latitude on Coupling between the Stratospheric Polar Vortex and the Troposphere, Journal of Climate, 26, doi:10.1175/JCLI-D-12-00301.1, 2013.

Garfinkel, C. I., Schwartz, C., Butler, A. H., Domeisen, D. I., Son, S.-W., and White, I. P.: Weakening of the teleconnection of El Nino-Southern Oscillation to the Arctic stratosphere over the past few decades: What can be learned from subseasonal forecast models, J. Geophys. Res.- Atmos., in press.

Hardiman, S. C., Butchart, N., Charlton-Perez, A. J., Shaw, T. A., Akiyoshi, H., Baumgaertner, A., Bekki, S., Braesicke, P., Chipperfield, M., Dameris, M., et al.: Improved predictability of the troposphere using stratospheric final warmings, Journal of Geophysical Research: Atmospheres, 116, 2011.

Hardiman, S. C., Butchart, N., O'Connor, F. M., and Rumbold, S. T.: The Met Office HadGEM3-ES chemistry–climate model: evaluation of stratospheric dynamics and its impact on ozone, Geoscientific Model Development, 10, 1209–1232, doi:10.5194/gmd-10-1209-2017, https://www.geosci-model-dev.net/10/1209/2017/, 2017.

Hartmann, D. L.: Some aspects of the coupling between radiation, chemistry, and dynamics in the stratosphere, Journal of Geophysical Research: Oceans, 86, 9631–9640, 1981.

Hartmann, D. L. and Garcia, R. R.: A mechanistic model of ozone transport by planetary waves in the stratosphere, Journal of the Atmospheric Sciences, 36, 350–364, 1979.

Hu, J., Li, T., Xu, H., and Yang, S.: Lessened response of boreal winter stratospheric polar vortex to El Niño in recent decades, Climate Dynamics, 49, 263–278, 2017.

Hurwitz, M. M., Newman, P. A., and Garfinkel, C. I.: The Arctic vortex in March 2011: a dynamical perspective, Atm. Chem. Phys., 11, 11 447–11 453, doi:10.5194/acp-11-11447-2011, 2011.

Ivy, D. J., Solomon, S., Calvo, N., and Thompson, D. W.: Observed connections of Arctic stratospheric ozone extremes to Northern Hemisphere surface climate, Environmental Research Letters, 12, 024 004, 2017.

Jöckel, P., Tost, H., Pozzer, A., Kunze, M., Kirner, O., Brenninkmeijer, C. A. M., Brinkop, S., Cai, D. S., Dyroff, C., Eckstein, J., Frank, F., Garny, H., Gottschaldt, K.-D., Graf, P., Grewe, V., Kerkweg, A., Kern, B., Matthes, S., Mertens, M., Meul, S., Neumaier, M., Nützel, M., Oberländer-Hayn, S., Ruhnke, R., Runde, T., Sander, R., Scharffe, D., and Zahn, A.: Earth System Chemistry integrated Modelling (ESCiMo) with the Modular Earth Submodel System (MESSy) version 2.51, Geoscientific Model Development, 9, 1153–1200, doi:10.5194/gmd-9-1153-2016, https://www.geosci-model-dev.net/9/1153/2016/, 2016.

Karpechko, A. Y., Perlwitz, J., and Manzini, E.: A model study of tropospheric impacts of the Arctic ozone depletion 2011, Journal of Geophysical Research: Atmospheres, 119, 7999–8014, 2014.

Kidston, J., Scaife, A. A., Hardiman, S. C., Mitchell, D. M., Butchart, N., Baldwin, M. P., and Gray, L. J.: Stratospheric influence on tropospheric jet streams, storm tracks and surface weather, Nature Geoscience, 8, 433–440, 2015.

Kirchner, I. and Peters, D.: Modelling the wintertime response to upper tropospheric and lower stratospheric ozone anomalies over the North Atlantic and Europe, in: Annales Geophysicae, vol. 21, pp. 2107–2118, 2003.

Li, J., Sun, C., and Jin, F.-F.: NAO implicated as a predictor of Northern Hemisphere mean temperature multidecadal variability, Geophysical research letters, 40, 5497–5502, 2013.

Li, Y., Li, J., and Feng, J.: A teleconnection between the reduction of rainfall in southwest Western Australia and north China, Journal of Climate, 25, 8444–8461, 2012.

Manney, G. L., Santee, M. L., Rex, M., Livesey, N. J., Pitts, M. C., Veefkind, P., Nash, E. R., Wohltmann, I., Lehmann, R., Froidevaux, L., et al.: Unprecedented Arctic ozone loss in 2011, Nature, 478, 469, 2011.

Meinshausen, M., Smith, S. J., Calvin, K., Daniel, J. S., Kainuma, M., Lamarque, J., Matsumoto, K., Montzka, S., Raper, S., Riahi, K., et al.: The RCP greenhouse gas concentrations and their extensions from 1765 to 2300, Climatic change, 109, 213–241, 2011.

Morgenstern, O., Braesicke, P., O'Connor, F. M., Bushell, A. C., Johnson, C. E., Osprey, S. M., and Pyle, J. A.: Evaluation of the new UKCA climate-composition model – Part 1: The stratosphere, Geoscientific Model Development, 2, 43–57, doi:10.5194/gmd-2-43-2009, https://www.geosci-model-dev.net/2/43/2009/, 2009.

Morgenstern, O., Akiyoshi, H., Bekki, S., Braesicke, P., Butchart, N., Chipperfield, M., Cugnet, D., Deushi, M., Dhomse, S., Garcia, R., et al.: Anthropogenic forcing of the Northern Annular Mode in CCMVal-2 models, Journal of Geophysical Research: Atmospheres, 115, 2010.

Morgenstern, O., Hegglin, M. I., Rozanov, E., O'Connor, F. M., Abraham, N. L., Akiyoshi, H., Archibald, A. T., Bekki, S., Butchart, N., Chipperfield, M. P., Deushi, M., Dhomse, S. S., Garcia, R. R., Hardiman, S. C., Horowitz, L. W., Jöckel, P., Josse, B., Kinnison, D., Lin, M., Mancini, E., Manyin, M. E., Marchand, M., Marécal, V., Michou, M., Oman, L. D., Pitari, G., Plummer, D. A., Revell, L. E., Saint-Martin, D., Schofield, R., Stenke, A., Stone, K., Sudo, K., Tanaka, T. Y., Tilmes, S., Yamashita, Y., Yoshida, K., and Zeng, G.: Review of the global models used within phase 1 of the Chemistry–Climate Model Initiative (CCMI), Geoscientific Model Development, 10, 639–671, doi:10.5194/gmd-10-639-2017, https://www.geosci-model-dev.net/10/639/2017/, 2017.

Newman, P., Daniel, J., Waugh, D., and Nash, E.: A new formulation of equivalent effective stratospheric chlorine (EESC), Atmospheric Chemistry and Physics, 7, 4537–4552, doi:10.5194/acp-7-4537-2007, 2007.

Pearl, J.: Causality: Models, Reasoning, and Inference, Cambridge University Press, New York, NY, USA, 2000.

Polvani, L. M., Waugh, D. W., Correa, G. J. P., and Son, S.-W.: Stratospheric Ozone Depletion: The Main Driver of Twentieth-Century Atmospheric Circulation Changes in the Southern Hemisphere, Journal of Climate, 24, 795–812, doi:10.1175/2010JCLI3772.1, 2011.

Pyper, B. J. and Peterman, R. M.: Comparison of methods to account for autocorrelation in correlation analyses of fish data, Canadian Journal of Fisheries and Aquatic Sciences, 55, 2127–2140, https://search.proquest.com/docview/219322992?accountid=14546, copyright - Copyright National Research Council of Canada Sep 1998; Document feature - Graphs; Tables; ; Last updated - 2014-05-21; CODEN - CJFSDX, 1998.

Rienecker, M. M., Suarez, M. J., Gelaro, R., Todling, R., Bacmeister, J., Liu, E., Bosilovich, M. G., Schubert, S. D., Takacs, L., Kim, G.-K., Bloom, S., Chen, J., Collins, D., Conaty, A., da Silva, A., Gu, W., Joiner, J., Koster, R. D., Lucchesi, R., Molod, A., Owens, T., Pawson, S., Pegion, P., Redder, C. R., Reichle, R., Robertson, F. R., Ruddick, A. G., Sienkiewicz, M., and Woollen, J.: MERRA: NASA's Modern-Era Retrospective Analysis for Research and Applications, Journal of Climate, 24, 3624–3648, doi:10.1175/JCLI-D-11-00015.1, 2011.

Rood, R. B. and Douglass, A. R.: Interpretation of ozone temperature correlations: 1. Theory, Journal of Geophysical Research: Atmospheres, 90, 5733–5743, 1985.

Runge, J., Petoukhov, V., Donges, J. F., Hlinka, J., Jajcay, N., Vejmelka, M., Hartman, D., Marwan, N., Paluš, M., and Kurths, J.: Identifying causal gateways and mediators in complex spatio-temporal systems, Nature communications, 6, 8502, 2015.

Runge, J., Sejdinovic, D., and Flaxman, S.: Detecting causal associations in large nonlinear time series datasets, arXiv preprint arXiv:1702.07007, 2017.

Samarasinghe, S., McGraw, M., Barnes, E., and Ebert-Uphoff, I.: A study of links between the Arctic and the midlatitude jet stream using Granger and Pearl causality, Environmetrics, p. e2540, 2018.

Silverman, V., Harnik, N., Matthes, K., Lubis, S. W., and Wahl, S.: Radiative effects of ozone waves on the Northern Hemisphere polar vortex and its modulation by the QBO, Atmospheric Chemistry and Physics Discussions, 2017, 1–33, doi:10.5194/acp-2017-641, https://www.atmos-chem-phys-discuss.net/acp-2017-641/, 2017.

Smith, K. L. and Polvani, L. M.: The surface impacts of Arctic stratospheric ozone anomalies, Environmental Research Letters, 9, 074 015, 2014.

SPARC-CCMVal: SPARC Report on the Evaluation of Chemistry-Climate Models, SPARC Report, 5, WCRP-132, WMO/TD-No. 1526, http://www.sparc-climate.org/publications/sparc-reports/sparc-report-no5/, 2010.

Spirtes, P. and Glymour, C.: An Algorithm for Fast Recovery of Sparse Causal Graphs, Social Science Computer Review, 9, 62–72, doi:10.1177/089443939100900106, https://doi.org/10.1177/089443939100900106, 1991.

Staehelin, J., Harris, N., Appenzeller, C., and Eberhard, J.: Ozone trends: A review, Reviews of Geophysics, 39, 231–290, 2001.

Tilmes, S., Lamarque, J.-F., Emmons, L. K., Kinnison, D. E., Marsh, D., Garcia, R. R., Smith, A. K., Neely, R. R., Conley, A., Vitt, F., Val Martin, M., Tanimoto, H., Simpson, I., Blake, D. R., and Blake, N.: Representation of the Community Earth System Model (CESM1) CAM4-chem within the Chemistry-Climate Model Initiative (CCMI), Geoscientific Model Development, 9, 1853–1890, doi:10.5194/gmd-9-1853-2016, https://www.geosci-model-dev.net/9/1853/2016/, 2016.

Vimont, D. J., Wallace, J. M., and Battisti, D. S.: The seasonal footprinting mechanism in the Pacific: Implications for ENSO, Journal of Climate, 16, 2668–2675, 2003.

Waugh, D. W., Garfinkel, C., and Polvani, L. M.: Drivers of the recent tropical expansion in the Southern Hemisphere: Changing SSTs or ozone depletion?, J. Clim., 28, 6581–6586, doi:10.1175/JCLI-D-15-0138.1, 2015.

World Meteorological Organization: Scientific Assessment of Ozone Depletion: 2010, Global Ozone Research and Monitoring Project Rep. No. 52, 2011.

World Meteorological Organization: Scientific Assessment of Ozone Depletion: 2014, Global Ozone Research and Monitoring Project Rep. No. 55, 2014.

World Meteorological Organization: Scientific Assessment of Ozone Depletion: 2018 Executive Summary, Global Ozone Research and Monitoring Project Rep. No. 58, 2018.

Xie, F., Li, J., Tian, W., FU, Q., Jin, F.-F., Hu, Y., Zhang, J., Wang, W., Sun, C., Feng, J., Yang, Y., and Ding, R.: A connection from Arctic stratospheric ozone to El Niño-Southern oscillation, Environmental Research Letters, 11, 124 026, 2016.

Yukimoto, S., Adachi, Y., Hosaka, M., Sakami, T., Yoshimura, H., Hirabara, M., Tanaka, T. Y., Shindo, E., Tsujino, H., Deushi, M., et al.: A new global climate model of the Meteorological Research Institute: MRI-CGCM3—model description and basic performance—, Journal of the Meteorological Society of Japan. Ser. II, 90, 23–64, 2012.