# Peer review of "Influence of Arctic Stratospheric Ozone on Surface Climate in CCMI models"

_Atmospheric Chemistry and Physics, 2018_

## Referee Comment (RC1) · Anonymous Referee #1 · 12 Nov 2018

The paper by Harari et al "Influence of Arctic Stratospheric Ozone on Surface Climate in CCMI models" investigates statistical links between zonal mean ozone averaged over the polar cap in the lower stratosphere and surface climate variables, such as surface temperature and sea level pressure, in seven coupled chemistry-climate models with interactive chemistry. The authors report statistically significant correlation between ozone and surface climate however they provide little discussion on mechanisms behind apparent links. Although the work may potentially lead to some interesting results I cannot recommend this current paper for publication for the reasons outlined below:

1. The motivation of the paper is not clear. The authors state that they "...revisit the connection between boreal spring Arctic stratospheric ozone variability on interannual timescales and surface climate..." but first of all I don't think there is any doubt that

[Figure]

ozone variability influences surface climate and I don't see why this needs to be re-visited. Ozone is radiatively active gas; it absorbs solar radiation in the stratosphere and thus heats the stratospheric air. This in turn affects stratospheric circulation and thereafter the troposphere and surface climate via stratosphere-troposphere dynamical coupling. Do the authors want to revisit this link? In any case I don't think the statistical approach adopted by the authors can provide a progress here.

Of more interest is the question of whether anthropogenic emission of ozone-depleted substances affected surface climate via affecting Arctic ozone. But I don't see that the authors address this question because they mix together periods when ozone was depleted (1970-2010), recovering (2011-2051) and fully recovered or possibly even "super-recovered" (2052-2092). Thus I think the authors need to reevaluate the motivation and objective of their study.

2. While reporting on statistical links the authors avoid discussing on possible mechanism. While it is true that inferring physical mechanisms from statistical relations is difficult, it would be valuable if the authors formulate clearer which mechanisms they keep in mind when they say "ozone influence on tropospheric climate". Do they mean (a) ozone induced dynamical changes in the stratosphere and following dynamical downward coupling or (b) downward radiative fluxes due to ozone variability, or something else? For example in the case of Antarctic ozone depletion, the likely mechanism through which ozone affects surface climate is through radiative cooling of the stratosphere and subsequent downward dynamical influence. Additionally, Grise et al (2009) studied possible radiative impacts of ozone depletion on the troposphere and found that most of the impacts is through cooling of the stratosphere leading to reduced downward flux of the infrared radiation. At the same time there is very little direct impacts of ozone on stratospheric transmissivity and emissivity. While this result appears in agreement with authors results according to which ozone connection to surface climate is "mediated by the dynamical variability" I don't think the authors provide new findings here.

[Figure]

3. On the basis of stronger statistical links between stratospheric dynamical indexes and surface climate, the authors conclude, "A connection between Arctic ozone variability and polar cap sea-level pressure is also found, but additional analysis suggests that it is mediated by the dynamical variability that typically drives the anomalous ozone concentrations." It is true that stratospheric transport determines ozone distribution but ozone also affects stratospheric circulation through radiative heating. I don't think author's analysis can rule out the possibility that stratospheric circulation that affected the surface climate was modified by ozone variability.

4. The authors report that they found "connection" between Arctic ozone variability and polar cap sea-level pressure and El Nino but looking at their results one can see correlation coefficients in the order of 0.1..0.2 in their multi-model mean results. While these may be statistically significant I really wonder about their usefulness especially since there is lack of physical understanding. It would be desirable if the authors proposed ways how these links can be utilized; however this has not been done.

In summary I am not sure if the authors can address the above issues within the current manuscript and therefore I recommend a rejection. Perhaps a good way forward is to adopt the approach by Calvo et al. and analyze periods of ozone depletion and ozone recovery separately trying to isolate the role of ozone rather than making an obvious (in my opinion) point that the ozone impact is mediated by the stratospheric dynamics.

Reference: Grise, K. M., D. W. J. Thompson, and P. M. Forster (2009), On the role of radiative processes in stratosphere–troposphere coupling, J. Clim., pp. 4154–4161, doi:10.1175/2009JCLI2756.1

Minor points:

P2L5: "This sensitivity suggests that the radiative perturbation due to ozone requires tropospheric feedback" Can it also be interpreted that ozone forcing in isolation is too weak to modify stratospheric circulation and that additional forcing from SST –driven wave activity is needed?

P3L12: "but utilizes up-to-date CCMs". Perhaps also including tropospheric chemistry?

P3L13: remove double "the"

P3L20: "in all cases there is a peak between 2 and 5 years (not shown)." In agreement with observations?

P3L28: Please explain how do you get 42 model samples?

Table 1: Add reanalysis to the table caption

P4L10: What does it mean: "limited data was either missing, corrupted, or non-physical"?

P7L7: When testing statistical hypotheses there could be only two outcomes, either the test is passed or not. There is no "nearly statistically significant" results. Based on your Figure 6 I conclude that the null hypothesis that the observed correlation between ASO and ENSO is accidental cannot be ruled out at the 5% rejection level.

P7L29: What do you mean by "the same model" here?

Figure 4: This figure makes me crazy. Correlation of correlation coefficients?!! What am I suppose to learn, for example, from the fact that there is a correlation of 0.54 between r(PS_March, ASO) and r(ZMArch_ASO)? I don't think the discussion in the text makes my task any easier. My specific concern is Figure 4d where the spread of the crosses is visually inconsistent with the reported correlation of 0.6 (for example the correlation of just 0.54 in the nearby Fig. 4c visually appears tighter). Can the authors double-check this number?

---

## Referee Comment (RC2) · Anonymous Referee #2 · 21 Nov 2018

Summary This paper evaluates relationships of Arctic stratospheric ozone (ASO) and surface climate, and ASO and El Niño-Southern Oscillation (ENSO), using model output from the Chemistry-Climate Model Initiative (CCMI). They find that the connection between ASO and surface climate may arise due to dynamic variability in the lower stratosphere, which has a strong correlation with both ASO and surface climate. They also find a weaker-than-observed (though "still significant") relationship of ASO and ENSO, with ASO leading ENSO by one to two years.

General Comments In general, I think the writing is clear and the analysis is well-explained. I think there is definitely utility in looking at these relationships in state-of-the-art chemistry-climate models, particularly since it's apparent that some of the observed relationships might be an artifact of sampling. My one major comment relates

to the relationship of ASO to ENSO. While the authors do due diligence and attempt to make sure this relationship isn't a factor of auto-correlation, I'm confused about their method. For example, I don't understand why the "minimal useful correlation" is basically zero (so basically anything greater than zero is useful?). See specific comments below too. I think this technique at a minimum needs to be better explained, but I would also recommend addressing the issue using Granger causality techniques, as in McGraw and Barnes (2018). Additionally, I think in parts the authors are attempting to use their results to support observational results in Xie et al. but I think if anything their results more clearly indicate the weakness of this relationship between ASO and ENSO, and that it's likely a sampling artifact in the observations. I would encourage them to remove statements such as in the abstract that state that ASO "may also influence the surface in both polar and tropical latitudes"- my impression from the results was actually the opposite, that ASO has very little influence on either polar or tropical surface climate, and that significant correlations can occur randomly for 40-year subsets, which is probably what we are seeing in the observations. In general I think understanding these possible relationships between Arctic ozone and surface climate is important, and the CCMI dataset provides a new tool to do so (particularly since it's hard to find models that are both coupled and have interactive chemistry and decent stratospheric processes). But I would like to see the authors address these concerns. I recommend a major revision.

Specific Comments Page 1, Line 16- maybe mention why ASO has been "spared from the worst ozone destruction", e.g., the relatively warmer polar temperatures due to stronger wave forcing.

Page 3, Line 12 – Can you explain what Ref-C2 is, i.e., what radiative forcings, specifications do these runs use.

Page 3, comment about Data- you need to include more information about SWOOSH- how long is the record, which version are you using, which data goes into SWOOSH, are you using the "anomaly filled" version, which latitude resolution, etc. One thing

I'd be interested to know- how much data does SWOOSH have polewards of 60N to calculate the ASO? Is there any data from 80-90N? If not, do you need to take that into account when comparing the model ozone?

Page 4, lines 5-7- is the multiple linear regression performed before or after subdivision into 40 year chunks? (does it matter?). Is this removal of the GHG/ODS effect done for both dynamic (T,Z) and ozone time series? Is this also performed for MERRA2/SWOOSH data? Page 6, line 5-7- change to "ASO and polar cap SLP than is observed in March" (the connection is stronger than observed in April). Also, is the statistical significance true for the r=0.09 value? Or just the April r=0.17 value? Even if it's significant... is a value like r=0.09 very useful? It's implying that only 0.8% of the variance in polar cap SLP is explained by ASO. Stating only that it's significant statistically may be misleading (and, I think, not strong support for statements in the abstract or conclusions that suggest such a relationship provides useful information about predictability).

Page 6, lines 24-27- might acknowledge here that there could be non-linear feedbacks at play that linear regression would not remove

Page 7, line 7- in general, I find the authors to be trying too hard in this section to reinforce Xie et al (2016) results; this statement is an example- "This relationship is nearly statistically significant at the 95% level". This should be changed to either the specific significance level that it meets, or it should say "this relationship is not significant at the 95% level". Particularly since, if anything is striking about Figure 6, it's that almost no correlations shown (not even the observed ones) meet significance levels. These results to me more strongly argue that the relationships suggested by Xie et al. are artificial. Page 8, line 29-30 more effectively state what I think the results of this study conclude- that there is no strong evidence of this mechanism/relationship in the CCMI models. But this message isn't clearly reflected in either the abstract or the conclusions (such as point (3) on page 11, line 1)

Page 7, Line 22- could this have to do with the power spectra of ENSO in models having higher amplitude at periods of 12 months instead of 24 months? (e.g., AchutaRao and Sperber 2002). Again, this would suggest these lead-lag relationships are more a reflection of ENSO auto-correlation than physically-based relationships.

Page 9, Line 1-10- I commend the authors for trying to deal with the auto-correlation issue. However, I'm not sure I understand their method. Why is ASO at a lag of 3 months chosen? Why not at the lag where the relationship peaks, in either the observations or the model? Then the "minimally useful correlation" is shown in equation (2) and plotted in red in Figure 6/7, but it's not clear to me what the right side of that equation has to do with the correlation values in plotted in black (yet it's then stated that "for both observations and the CCMI models the actual correlation between ASO and ENSO far-exceeds the minimally useful one"... but the actual correlation r(ENSO,ASO) is not part of the criteria in equation 2). Also confusing is that the minimally useful correlation appears to be nearly zero at all lags, so how is the criteria in eqn(2) satisfied? This should be clarified to better explain what this analysis tells us, but I also would recommend that instead of this method, additionally consider applying the Granger causality techniques as detailed in McGraw and Barnes (2018).

Page 9, line 13- this relationship at zero lag is true only in the multi-model mean CCMI, right? It seems to be the opposite sign in the observations.

Page 10, lines 1-17- this part of the conclusions was nicely written and well-phrased.

Technical Corrections Page 3, Line 6- Capitalize the appropriate letters for MERRA

Page 3, Line 18- remove repeated "the"

Page 5, Line 8- remove comma after heights and put it instead after the first word "stratosphere".

Page 5, Line 17-18- should be "with a single x, and the...". Also it should be yellow asterisk, not green?

Page 5, Line 27- remove "polar cap" and change to "sea level pressure anomalies" (I assume climatology is removed?). Could also add "anomalies" in line 31 (and elsewhere throughout paper). Figure 5a shows the correlation of sea level pressure anomalies at each grid point with the ASO in March, right?

Page 5, Line 30- semi-colon instead of comma after Ivy et al. 2017.

Page 9, line 5- should be "3 months"

Page 10, line 10- change "an" to "a"

Page 11, line 5- capitalize "acknowledgement"

Page 14, caption- misspelled "stratosphere" on line 3. Should these be stated as anomalies in T and Z or are these full fields? (also true in other captions) Also the y-axis on panel (h) seem mislabeled (should be ZApr?). Note also in y-axis of (a) and (b) that the correlation value is 0.36 on one and 0.35 in the other; I believe this should be the same number.

---

## Short Comment (SC1) · 2 Dec 2018

The paper by Harari et al "Influence of Arctic Stratospheric Ozone on Surface Climate in CCMI models" investigates statistical links between zonal mean ozone averaged over the polar cap in the lower stratosphere and surface climate variables, such as surface temperature and sea level pressure, in seven coupled chemistry-climate models with interactive chemistry. The authors report statistically significant correlation between ozone and surface climate however they provide little discussion on mechanisms behind apparent links. Although the work may potentially lead to some interesting results I cannot recommend this current paper for publication for the reasons outlined below:

1. The motivation of the paper is not clear. The authors state that they ". . .revisit the connection between boreal spring Arctic stratospheric ozone variability on interannual timescales and surface climate. . ." but first of all I don't think there is any doubt that ozone variability influences surface climate and I don't see why this needs to be revisited. Ozone is radiatively active gas; it absorbs solar radiation in the stratosphere and thus heats the stratospheric air. This in turn affects stratospheric circulation and thereafter the troposphere and surface climate via stratosphere-troposphere dynamical coupling. Do the authors want to revisit this link? In any case I don't think the statistical approach adopted by the authors can provide a progress here.

Of more interest is the question of whether anthropogenic emission of ozone-depleted substances affected surface climate via affecting Arctic ozone. But I don't see that the authors address this question because they mix together periods when ozone was depleted (1970-2010), recovering (2011-2051) and fully recovered or possibly even "super-recovered" (2052-2092). Thus I think the authors need to reevaluate the moti-vation and objective of their study.

The reviewer appears to believe that the existence of a connection between Arctic ozone and surface climate is settled science. As discussed in the introduction (but evidently not clearly enough), three studies by three different research groups have reached the opposite conclusion. We are aware of only one modeling study that attempted to address this question that has reached the conclusion that there is a robust impact.

The 2018 WMO ozone assessment includes a review of this issue, and consistent with the introduction of our paper, the ozone assessment concludes that "interannual variability in springtime Antarctic and Arctic ozone may be important for surface climate, but work remains to better quantify this connection." and also that "future work is needed to evaluate whether differences in the ozone forcings, as well as other inter-model differences, among the various studies has contributed to the range of conclusions." The ozone assessment is the closest document the ozone community produces that is intended to reflect the current scientific consensus, and it clearly indicates that this issue is not settled and deserves revisiting. These statements are quoted from the fifth order draft from July 2018, i.e. after all scientific reviews had already been completed.

That being said, we plan to separately consider interannual variability in the depleted, recovering, and super-recovered states in our revised manuscript, but our preliminary analysis indicates that all results in this paper are valid if one considers just the historical depleted period, and differences among the three periods are minor (though see the figures in response to comment 3 below).

2. While reporting on statistical links the authors avoid discussing on possible mechanism. While it is true that inferring physical mechanisms from statistical relations is difficult, it would be valuable if the authors formulate clearer which mechanisms they keep in mind when they say "ozone influence on tropospheric climate". Do they mean (a) ozone induced dynamical changes in the stratosphere and following dynam-ical downward coupling or (b) downward radiative fluxes due to ozone variability, or something else? For example in the case of Antarctic ozone depletion, the likely mechanism through which ozone affects surface climate is through radiative cooling of the stratosphere and subsequent downward dynamical influence. Additionally, Grise et al (2009) studied possible radiative impacts of ozone depletion on the troposphere and found that most of the impacts is through cooling of the stratosphere leading to reduced downward flux of the infrared radiation. At the same time there is very little direct impacts of ozone on stratospheric transmissivity and emissivity. While this result appears in agreement with authors results according to which ozone connection to surface climate is "mediated by the dynamical variability" I don't think the authors provide new findings here.

The goal of this manuscript was not to shed light on mechanisms in these CCMI models. Rather the novelty of revisiting the Arctic Ozone-surface climate connection in CCMI models is that these models have an interactive ocean (which wasn't true of previous model generations), and also we have more than 1600 years of model output we can use to put the observed results in context (in contrast to previous modeling studies which had an order of magnitude less data available). As discussed in Garfinkel et al 2013 and Garfinkel and Waugh 2014, it is very difficult to tease out mechanisms as to how downward coupling occurs from long model runs like those contributed to the CCMI project. The best we can do is to form "emergent constraints", that is to use the connection between e.g. ozone and stratospheric conditions to better understand how ozone may affect surface climate. That being said, we plan to discuss this issue in more detail in the revised manuscript including a better description of our figures comparing correlations to correlations, and also to better explain the novel aspects of our analysis.

(As an aside, the conclusion of Grise et al 2009 is that the radiative mechanism the reviewer has in mind isn't important with regards to Arctic ozone.)

3.
On the basis of stronger statistical links between stratospheric dynamical indexes and surface climate, the authors conclude, "A connection between Arctic ozone vari-ability and polar cap sea-level pressure is also found, but additional analysis suggests that it is

mediated by the dynamical variability that typically drives the anomalous ozone concentrations." It is true that stratospheric transport determines ozone distribution but ozone also affects stratospheric circulation through radiative heating. I don't think author's analysis can rule out the possibility that stratospheric circulation that affected the surface climate was modified by ozone variability.

We certainly agree that the stratospheric circulation anomalies that affected the surface could have modified by ozone variability, and we never meant to imply the contrary.

Below are two figures we have produced in which the x-axis shows the correlation between polar cap temperature at 100hPa and polar cap SLP for each 40 year subsample of each model, and the y-axis shows the correlation between polar cap geopotential height at 100hPa and polar cap SLP for each 40 year subsample of each model. Subsamples during the "super-recovery" period are in red, during the recovery period in green, and during the depleted period in blue. The top panel is for February and the bottom panel is for March. The mean of each period is indicated with a square. The connection between conditions in the stratosphere and surface climate is stronger during the super-recovery period by up to a factor of two. We plan to include the figures below in the revised manuscript.

We realize that the sentence highlighted by the reviewer was poorly phrased, and we plan to clarify it for the revised manuscript. However our results do indicate that the dynamical pathway is dominant, as if we statistically remove the dynamical pathway then the connection between ozone and subpolar surface climate goes away.

[Figure]

[Figure]

4. The authors report that they found "connection" between Arctic ozone variability and polar cap sea-level pressure and El Nino but looking at their results one can see correlation coefficients in the order of 0.1..0.2 in their multi-model mean results. While these

may be statistically significant I really wonder about their usefulness especially since there is lack of physical understanding. It would be desirable if the authors proposed ways how these links can be utilized; however this has not been done.

We never meant to imply that a correlation on the order of 0.1 is useful in an operational sense. Correlations in observational data are almost four times higher, however, and hence indicate that such a relationship could be useful. Our goal was to assess whether this observational relationship is simulated by the CCMI models (the first and only multimodel ensemble that could be reasonably expected to capture this relationship), and more crucially to evaluate the spread in response in the models in order to get a sense of whether the observed response is "forced" or just reflects internal variability in a short sample. The fact that the multi-model mean signal is ~4 times weaker than the observed signal, but that individual 40-year subsamples show relationships very similar to that observed, indicate that the borderline useful observational result is potentially inflated by internal variability. We plan to clarify this issue in the revised manuscript.

In summary I am not sure if the authors can address the above issues within the current manuscript and therefore I recommend a rejection. Perhaps a good way forward is to adopt the approach by Calvo et al. and analyze periods of ozone depletion and ozone recovery separately trying to isolate the role of ozone rather than making an obvious (in my opinion) point that the ozone impact is mediated by the stratospheric dynamics.

As noted above, with careful rewriting and some new analysis that has already been completed we believe we can address those comments of the reviewer that are reasonable (i.e. the second, third, and fourth comments). More specifically, we plan to include figures analogous to what is shown in the response to comment 3 above in the revised manuscript. With one notable exception (that shown above), there is generally little difference between the super-recovery period and the depleted period. We also plan to address the minor comments below in the revised manuscript.

Reference: Grise, K. M., D. W. J. Thompson, and P. M. Forster (2009), On the role of radiative processes in stratosphereâA Rtroposphere  coupling, J. Clim., pp. 4154–4161, doi:10.1175/2009JCLI2756.1

Minor points:

P2L5: "This sensitivity suggests that the radiative perturbation due to ozone requires tropospheric feedback" Can it also be interpreted that ozone forcing in isolation is too weak to modify stratospheric circulation and that additional forcing from SST –driven wave activity is needed?

P3L12: "but utilizes up-to-date CCMs". Perhaps also including tropospheric chemistry?

P3L13: remove double "the"

P3L20: "in all cases there is a peak between 2 and 5 years (not shown)." In agreement with observations?

P3L28: Please explain how do you get 42 model samples?

Table 1: Add reanalysis to the table caption

P4L10: What does it mean: "limited data was either missing, corrupted, or non-physical"?

P7L7: When testing statistical hypotheses there could be only two outcomes, either the test is passed or not. There is no "nearly statistically significant" results. Based on your Figure 6 I conclude that the null hypothesis that the observed correlation between ASO and ENSO is accidental cannot be ruled out at the 5% rejection level.

P7L29: What do you mean by "the same model" here?

Figure 4: This figure makes me crazy. Correlation of correlation coefficients?!! What am I suppose to learn, for example, from the fact that there is a correlation of 0.54 between r(PS_March, ASO) and r(ZMArch_ASO)? I don't think the discussion in the text makes my task any easier. My specific concern is Figure 4d where the spread of the crosses is visually inconsistent with the reported correlation of 0.6 (for example the correlation of just 0.54 in the nearby Fig. 4c visually appears tighter). Can the authors double-check this number?

---

## Author Comment (AC1) · 29 Jan 2019

Summary This paper evaluates relationships of Arctic stratospheric ozone (ASO) and surface climate, and ASO and El Niño-Southern Oscillation (ENSO), using model out- put from the Chemistry-Climate Model Initiative (CCMI). They find that the connection between ASO and surface climate may arise due to dynamic variability in the lower stratosphere, which has a strong correlation with both ASO and surface climate. They also find a weaker-than-observed (though "still significant") relationship of ASO and ENSO, with ASO leading ENSO by one to two years.

General Comments In general, I think the writing is clear and the analysis is well- explained. I think there is definitely utility in looking at these relationships in state-of- the-art chemistry-climate models, particularly since it's apparent that some of the ob- served relationships might be an artifact of sampling. My one major comment relates
to the relationship of ASO to ENSO. While the authors do due diligence and attempt to make sure this relationship isn't a factor of auto-correlation, I'm confused about their method. For example, I don't understand why the "minimal useful correlation" is basi- cally zero (so basically anything greater than zero is useful?). See specific comments below too. I think this technique at a minimum needs to be better explained, but I would also recommend addressing the issue using Granger causality techniques, as in Mc- Graw and Barnes (2018).

Thank you for your helpful and constructive suggestion. We have removed the minimally useful correlation and replaced it with a new analysis based on causal effect networks, as suggested by the reviewer. This rather new technique is methodically outlined and implemented in Runge et al 2017 on a climate dataset. The specific technique we use is based on Pearl causality (Pearl 2009) and is somewhat different from Granger causality techniques (i.e, Mcgraw and Barnes (2018)). The benefits and drawbacks of Granger vs Pearl causality are discussed in Runge et al 2017. Note that Pearl causality has been used in climate science by Kretschmer et al 2016.

The causal effect networks analysis is based on a two-step algorithm. The first step is the PC algorithm (Spirtes and Glymour, 1991). This step is used to find the "parents" (i.e. of a time-series) while the next step is used to quantify the causal

strength of the first step. The full analysis description is outlined in the revised manuscript. In short, we produced three variables, each one is a time-series (ASO, ENSO and Zpole), from the SWOOSH and CCMI models. The PC step of the analysis was used to find the parents of each variable, within a lag of 10 to 27 months prior to it. The significance threshold that was used is 0.05 (note that this threshold has a different statistical interpretation than that used in e.g. Student t tests- see Runge et.al 2017). In contrast to Runge et al 2017, we use the PC step with $q_{max}=10$ (maximum combinations of conditions) as the original PC algorithm suggests. In the second step of the analysis we used two different methods to evaluate the parents' causality strength. One method, named Partial Correlation, is used to calculate the correlation between two sets of residuals – that of a variable and that of its parent. The residuals are obtained by regressing out the influence of all other parents identified from the PC step (i.e. the first step). (see ParCorr alg. In Runge 2017). The partial correlation result was tested with a two tailed T test with alpha=0.05. The second method that is used to quantify the causal strength is called linear mediation (Runge 2015).

While this method can be used to compute different causal strength scores, we used it to calculate the beta coefficients of a multiple linear regression with the parents of the variable (e.g., ENSO) as regressors. The results for the ENSO variable are shown in the figure below copied from the revised manuscript:

[Figure]

Fig12: Results of the (a) PCMCI analysis and (b) PC algorithm with liner mediation. Both started by finding ENSO's Parents using the PC algorithm. Then, two methods were used to estimate the connection's strength: (a) partial correlation with 95% confidence level, and (b) computing the beta coefficients of ENSO's parents in the different time periods.

In observations/SWOOSH, ASO is a robust parent of ENSO at lags -20 and -22 months. However, for the historical period in the models, ASO is a robust parent of ENSO only for a few selected models but inconsistent in sign as compared to SWOOSH. In the periods of 2011-2051 and 2052-2092 more models show ASO as a parent of ENSO, from lags -10 to -27 months, with more sign consistency as compared to SWOOSH. The main parent of ENSO in the models is ENSO -10 months (i.e. auto-correlated), but not for observations (SWOOSH).

Runge J, Sejdinovic D, Flaxman S. Detecting causal associations in large nonlinear time series datasets. arXiv preprint arXiv:1702.07007. 2017 Feb 22.

( https://arxiv.org/abs/1702.07007 )

Runge et al. (2015): Identifying causal gateways and mediators in complex spatio-temporal systems. Nature Communications, 6, 8502. http://doi.org/10.1038/ncomms9502

Pearl, J., Causality: Models, Reasoning, and Inference (Cambridge University Press, Cambridge, 2009, 2nd edition)

Kretschmer, M., Coumou, D., Donges, J. F., & Runge, J. (2016). Using causal effect networks to analyze different Arctic drivers of midlatitude winter circulation. Journal of Climate, 29(11), 4069-4081.

P. Spirtes, C. Glymour, An Algorithm for Fast Recovery of Sparse Causal Graphs. Soc. Sci. Comput. Rev. 9, 62–72 (1991).

Additionally, I think in parts the authors are attempting to use their results to support observational results in Xie et al. but I think if anything their results more clearly indicate the weakness of this relationship between ASO and ENSO, and that it's likely a sampling artifact in the observations. I would encourage them to remove statements such as in the abstract that state that ASO "may also influence the surface in both polar and tropical latitudes"- my impression from the results was actu- ally the opposite, that ASO has very little influence on either polar or tropical surface climate, and that significant correlations can occur randomly for 40-year subsets, which is probably what we are seeing in the observations.

As will be discussed in our new discussion on causality, the observed apparent connection between ASO and ENSO is causal within the framework of Pearl causality, though we agree it is very weak. It is even weaker in the models. We suspect that this weak effect may not be particularly useful in an operational sense, and we plan to discuss this in the revised text.

In general, I think understanding these possible relationships between Arctic ozone and surface climate is important, and the CCMI dataset provides a new tool to do so (particularly since it's hard to find models that are both coupled and have interactive chemistry and decent stratospheric processes). But I would like to see the authors address these concerns. I recommend a major revision.

Specific Comments Page 1, Line 16- maybe mention why ASO has been "spared from the worst ozone destruction", e.g., the relatively warmer polar temperatures due to stronger wave forcing.

We have added " due to the relatively stronger wave forcing from the troposphere"

Page 3, Line 12 – Can you explain what Ref-C2 is, i.e., what radiative forcings, specifi- cations do these runs use.

Full details of the Ref-C2 simulations are described in Eyring et al. (2013); briefly, these simulations span the period 1960–2100, impose ozone depleting substances as in (World Meteorological Organization, 2011), and impose greenhouse gases other than ozone depleting substances as in RCP 6.0 (Meinshausen et al., 2011).

Page 3, comment about Data- you need to include more information about SWOOSH- how long is the record, which version are you using, which data goes into SWOOSH, are you using the "anomaly filled" version, which latitude resolution, etc. One thing I'd be interested to know- how much data does SWOOSH have polewards of 60N to calculate the ASO? Is there any data from 80-90N? If not, do you need to take that into account when comparing the model ozone?

For SWOOSH, we now define ASO as an area weighted mean ozone from 60-degree N to 81.25-degree N and mass-weighted average from 150hPa to 50hPa. The poleward limit of the region used to define ASO is set at 81.25N to match the data available from SWOOSH. We now also clarify that we use the combinedeqfillanomfillo3q product at 2.5-degree resolution with 31 vertical levels, and focus on the period 1984-2014.

Page 4, lines 5-7- is the multiple linear regression performed before or after sub- division into 40 year chunks?  (does it

matter?). Is this removal of the GHG/ODS effect done for both dynamic (T,Z) and ozone time series? Is this also performed for MERRA2/SWOOSH data?

We perform the MLR before dividing into 40-year chunks, now clarified. It is done for both dynamic and ozone time series, also now clarified. It is also applied to MERRA/SWOOSH, also clarified.

Page 6, line 5-7- change to "ASO and polar cap SLP than is observed in March" (the connection is stronger than observed in April).

Changed as suggested

Also, is the statistical significance true for the r=0.09 value? Or just the April r=0.17 value? Even if it's significant. . . is a value like r=0.09 very useful? It's implying that only 0.8% of the variance in polar cap SLP is explained by ASO. Stating only that it's significant statistically may be misleading (and, I think, not strong support for statements in the abstract or conclusions that suggest such a relationship provides useful information about predictability).

Even the 0.09 correlation is significant due to the >1600 models years available. However, we agree that while these correlations are statistically significant at the 95% level, the variance explained is low and hence ASO may not be particularly useful for prediction of surface climate. This has been clarified.

Page 6, lines 24-27- might acknowledge here that there could be non-linear feedbacks at play that linear regression would not remove

We agree. We have added a new figure where we compute the correlation between polar cap height at 100hPa and polar cap SLP from 1970 to 2010, from 2011-2051, and from 2052-2092 We do indeed find differences among these three periods. The accompanying discussion for this figure highlights that while the linear relationship between ozone and polar cap SLP is indistinguishable from that associated with polar cap height, there is certainly the possibility for nonlinear feedbacks.

Page 7, line 7- in general, I find the authors to be trying too hard in this section to rein- force Xie et al (2016) results; this statement is an example- "This relationship is nearly statistically significant at the 95% level". This should be changed to either the specific significance level that it meets, or it should say "this relationship is not significant at the 95% level". Particularly since, if anything is striking about Figure 6, it's that almost no correlations shown (not even the observed ones) meet significance levels. These results to me more strongly argue that the relationships suggested by Xie et al. are artificial.

We now state that these correlations are not statistically significant at the 95% level.

Xie et al in contrast claims that it is significant, and we now note that there is a difference in the level of significance between our results and those of Xie et al.

Page 8, line 29-30 more effectively state what I think the results of this study conclude- that there is no strong evidence of this mechanism/relationship in the CCMI models. But this message isn't clearly reflected in either the abstract or the conclusions (such as point (3) on page 11, line 1)

When looking at the CCMI models from 1970-2010, we do not see strong evidence of the relationship between ASO and ENSO nor to the mechanism suggested by Xie et al. The observational data (SWOOSH 1984-2014), however does seems to be compatible with their results. In addition, for the CCMI models for the year 2011 and on, the prediction power of ENSO based on ASO seems somewhat larger. Hence while the relationship between ASO and ENSO has not yet been proven to actually be helpful in an operational sense, the relationship does appear to exist.

The revised version of the text will include a better discussion of this issue in the abstract and conclusions.

Page 7, Line 22- could this have to do with the power spectra of ENSO in models having higher amplitude at periods of 12 months instead of 24 months? (e.g., AchutaRao and Sperber 2002). Again, this would suggest these lead-lag relationships are more a reflection of ENSO auto-correlation than physically-based relationships.

The causality argument now takes into account ENSO autocorrelation explicity. The models indeed do not show any causal influence, though the observations do.

Page 9, Line 1-10- I commend the authors for trying to deal with the auto-correlation is- sue. However, I'm not sure I understand their method. Why is ASO at a lag of 3 months chosen? Why not at the lag where the relationship peaks, in either the observations or the model? Then the "minimally useful correlation" is shown in equation (2) and plotted in red in Figure 6/7, but it's not clear to me what the right side of that equation has to do with the correlation values in plotted in black (yet it's then stated that "for both observations and the CCMI models the actual correlation between ASO and ENSO far-exceeds the minimally useful one". . . but the actual correlation r(ENSO,ASO) is not part of the criteria in equation 2). Also confusing is that the minimally useful correlation appears to be nearly zero at all lags, so how is the criteria in eqn(2) satisfied? This should be clarified to better explain what this analysis tells us, but I also would recom- mend that instead of this method, additionally consider applying the Granger causality techniques as detailed in McGraw and Barnes (2018).

We have accepted the reviewer's suggestion, and now use causality techniques. The minimally useful correlation arguments have been removed.

Page 9, line 13- this relationship at zero lag is true only in the multi-model mean CCMI, right? It seems to be the opposite sign in

the observations.

Over the period of 1984 through 2014, the correlation of ENSO with polar cap geopotential height was indeed quite weak. See Hu et al 2017 and Domeisen et al 2019. We now note this apparent decadal variability.

Domeisen, D. I., Garfinkel, C. I., & Butler, A. H. The Teleconnection of El Niño Southern Oscillation to the Stratosphere. *Reviews of Geophysics.*

Hu, J., Li, T., Xu, H., & Yang, S. (2017). Lessened response of boreal winter stratospheric polar vortex to El Niño in recent decades. *Climate Dynamics*, *49*(1-2), 263-278.

Page 10, lines 1-17- this part of the conclusions was nicely written and well-phrased.

Thank you!

Technical Corrections Page 3, Line 6- Capitalize the appropriate letters for MERRA

fixed

Page 3, Line 18- remove repeated "the"

Fixed

Page 5, Line 8- remove comma after heights and put it instead after the first word "stratosphere".

fixed

Page 5, Line 17-18- should be "with a single x, and the. . .". Also it should be yellow asterisk, not green?
Fixed

Page 5, Line 27- remove "polar cap" and change to "sea level pressure anomalies" (I assume climatology is removed?). Could also add "anomalies" in line 31 (and else- where throughout paper). Figure 5a shows the correlation of sea level pressure anoma- lies at each grid point with the ASO in March, right?

fixed

Page 5, Line 30- semi-colon instead of comma after

Ivy et al. 2017.

fixed

Page 9, line 5- should be "3 months"

fixed

Page 10, line 10- change "an" to "a"
fixed

Page 11, line 5- capitalize "acknowledgement"

fixed

Page 14, caption- misspelled "stratosphere" on line 3.

fixed

Should these be stated as anomalies in T and Z or are these full fields? (also true in other captions)

Fixed

Also the y-axis on panel (h) seem mislabeled (should be ZApr?).

fixed

Note also in y-axis of (a) and (b) that the correlation value is 0.36 on one and 0.35 in the other; I believe this should be the same number.

It is not the same number – one is for March the other for April. In any event we have removed this from the figure, and now include a large black X to represent the multi-model mean.

C

---

## Author Comment (AC2) · 29 Jan 2019

The paper by Harari et al "Influence of Arctic Stratospheric Ozone on Surface Climate in CCMI models" investigates statistical links between zonal mean ozone averaged over the polar cap in the lower stratosphere and surface climate variables, such as surface temperature and sea level pressure, in seven coupled chemistry-climate models with interactive chemistry. The authors report statistically significant correlation between ozone and surface climate however they provide little discussion on mechanisms behind apparent links. Although the work may potentially lead to some interesting results I cannot recommend this current paper for publication for the reasons outlined below:

1. The motivation of the paper is not clear. The authors state that they ". . .revisit the connection between boreal spring Arctic stratospheric ozone variability on interannual timescales and surface climate. . ." but first of all I don't think there is any doubt that ozone variability influences surface climate and I don't see why this needs to be revisited. Ozone is radiatively active gas; it absorbs solar radiation in the stratosphere and thus heats the stratospheric air. This in turn affects stratospheric circulation and thereafter the troposphere and surface climate via stratosphere-troposphere dynamical coupling. Do the authors want to revisit this link? In any case I don't think the statistical approach adopted by the authors can provide a progress here.

Of more interest is the question of whether anthropogenic emission of ozone-depleted substances affected surface climate via affecting Arctic ozone. But I don't see that the authors address this question because they mix together periods when ozone was depleted (1970-2010), recovering (2011-2051) and fully recovered or possibly even "super-recovered" (2052-2092). Thus I think the authors need to reevaluate the moti-vation and objective of their study.

The reviewer appears to believe that the existence of a connection between Arctic ozone and surface climate is settled science. As discussed in the introduction (but not clearly enough), at least three different research groups have reached the opposite conclusion. We are aware of only one modeling study that attempted to address this question that has reached the conclusion that there is a robust impact, in addition to the observational study that also concludes there is an impact.

The 2018 WMO ozone assessment is the closest document the ozone community produces that is intended to reflect the current scientific consensus, and it clearly indicates that this issue is not settled and deserves revisiting. The Executive Summary appendix includes the statement "there are indications that occurrences of extremely low springtime ozone amounts in the Arctic may have short-term effects on Northern Hemisphere regional surface climate". This statement is consistent with the divergent conclusions of the five studies mentioned in our introduction. We have added this statement from the most recent assessment to the introduction.

The full report has not yet been published, but it should be coming out very soon. The fifth order draft from July 2018, i.e. after all scientific reviews had already been completed, includes statements such as "interannual variability in springtime Antarctic and Arctic ozone may be important for surface climate, but work remains to better quantify this connection." and also that "future work is needed to evaluate whether differences in the ozone forcings, as well as other inter-model differences, among the various studies has contributed to the range of conclusions."

That being said, we will separately consider interannual variability in the depleted, recovering, and super-recovered states in our revised manuscript, but our analysis indicates that all results in this paper are valid if one considers just the historical depleted period, and differences among the three periods are minor (though see the figures in response to comment 3 below).

2. While reporting on statistical links the authors avoid discussing on possible mechanism. While it is true that inferring physical mechanisms from statistical relations is difficult, it would be valuable if the authors formulate clearer which mechanisms they keep in mind when they say "ozone influence on tropospheric climate". Do they mean (a) ozone induced dynamical changes in the stratosphere and following dynam-ical downward coupling or (b) downward radiative fluxes due to ozone variability, or something else? For example in the case of Antarctic ozone depletion, the likely mechanism through which ozone affects surface climate is through radiative cooling of the stratosphere and subsequent downward dynamical influence. Additionally, Grise et al (2009) studied possible radiative impacts of ozone depletion on the troposphere and found that most of the impacts is through cooling of the stratosphere leading to reduced downward flux of the infrared radiation. At the same time there is very little direct impacts of ozone on stratospheric transmissivity and emissivity. While this result appears in agreement with authors results according to which ozone connection to surface climate is "mediated by the dynamical variability" I don't think the authors provide new findings here.

The goal of this manuscript was not to shed light on mechanisms in these CCMI models. Rather the novelty of revisiting the Arctic Ozone-surface climate connection in CCMI models is that these models have an interactive ocean (which wasn't true of previous model generations), and also we have more than 1600 years of model output we can use to put the observed results in context (in contrast to previous modeling studies which had an order of magnitude less data available). We now highlight these two novel aspects in both the introduction and conclusion.

We have added to the conclusions that "The mechanism whereby polar stratospheric variability influences the tropospheric circulation is beyond the scope of this work, though we suspect that it will be very difficult to tease out mechanisms from the CCMI models due to

tropospheric feedbacks reinforcing any initial response forced by the stratosphere (Garfinkel et al 2013, Garfinkel and Waugh 2014, Kidston et al 2015)"

(As an aside, the conclusion of Grise et al 2009 is that the radiative mechanism the reviewer has in mind isn't important with regards to Arctic ozone.)

3.

On the basis of stronger statistical links between stratospheric dynamical indexes and surface climate, the authors conclude, "A connection between Arctic ozone vari-ability and polar cap sea-level pressure is also found, but additional analysis suggests that it is mediated by the dynamical variability that typically drives the anomalous ozone concentrations." It is true that stratospheric transport determines ozone distribution but ozone also affects stratospheric circulation through radiative heating. I don't think au-thor's analysis can rule out the possibility that stratospheric circulation that affected the surface climate was modified by ozone variability.

We certainly agree that the stratospheric circulation anomalies that affected the surface could have modified by ozone variability, and we never meant to imply the contrary.

We realize that the sentence highlighted by the reviewer was poorly phrased, and we have clarified it for the revised manuscript. However our results do indicate that the dynamical pathway is dominant, as if we statistically remove the dynamical pathway then the connection between ozone and subpolar surface climate goes away.

In addition to the statistical arguments in the original submission, we have performed additional analysis to try to tease out whether ozone may be important for downward coupling. Below are two figures we have produced in which the x-axis shows the correlation between polar cap temperature at 100hPa and polar cap SLP for each 40 year subsample of each model, and the y-axis shows the correlation between polar cap geopotential height at 100hPa and polar cap SLP for each 40 year subsample of each model. Subsamples during the "super-recovery" period are in red, during the recovery period in green, and during the depleted period in blue. The top panel is for February and the bottom panel is for March. The mean of each period is indicated with a square. The connection between conditions in the stratosphere and surface climate is stronger during the super-recovery period by up to a factor of two. We plan to include the figures below in the revised manuscript.

[Figure]

[Figure]

| ✕ 1970–2010 | ✕ 2011–2051 | ✕ 2052–2092 | ● MERRA/SWOOSH |

4. The authors report that they found "connection" between Arctic ozone variability and polar cap sea-level pressure and El Nino but looking at their results one can see correlation coefficients in the order of 0.1..0.2 in their multi-model mean results. While these may be statistically significant I really wonder about their usefulness especially since there is lack of physical understanding. It would be desirable if the authors proposed ways how these links can be utilized; however this has not been done.

We never meant to imply that a correlation on the order of 0.1 is useful in an operational sense. Correlations in observational data are almost four times higher, however, and hence indicate that such a relationship could be useful. Our goal was to assess whether this observational relationship is simulated by the CCMI models (the first and only multimodel ensemble that could be reasonably expected to capture this relationship), and more crucially to evaluate the spread in response in the models in order to get a sense of whether the observed response is "forced" or just reflects internal variability in a short sample. The fact that the multi-model mean signal is ~4 times weaker than the observed signal, but that individual 40-year subsamples show relationships very similar to that observed, indicate that the borderline useful observational result is potentially inflated by internal variability. We plan to clarify this issue in the revised manuscript.

In summary I am not sure if the authors can address the above issues within the current manuscript and therefore I recommend a rejection. Perhaps a good way forward is to adopt the approach by Calvo et al. and analyze periods of ozone depletion and ozone recovery separately trying to isolate the role of ozone rather than making an obvious (in my opinion) point that the ozone impact is mediated by the stratospheric dynamics.

As noted above, with careful rewriting and some new analysis that has already been completed we can address those comments of the reviewer that are reasonable (i.e. the second, third, and fourth comments). More specifically, we plan to include figures analogous to what is shown in the response to comment 3

above in the revised manuscript. With one notable exception (that shown above), there is generally little difference between the super-recovery period and the depleted period. We also plan to address the minor comments below in the revised manuscript.

Reference: Grise, K. M., D. W. J. Thompson, and P. M. Forster (2009), On the role of radiative processes in stratosphereâA Rtroposphere coupling, J. Clim., pp. 4154–4161, doi:10.1175/2009JCLI2756.1

Minor points:

P2L5: "This sensitivity suggests that the radiative perturbation due to ozone requires tropospheric feedback" Can it also be interpreted that ozone forcing in isolation is too weak to modify stratospheric circulation and that additional forcing from SST –driven wave activity is needed?

The Karpechko et al paper does indeed conclude that the stratospheric response is too weak if ozone is forced in isolation. The reduced upward wave flux in 2011 was also important. This has been clarified.

P3L12: "but utilizes up-to-date CCMs". Perhaps also including tropospheric chemistry?

We now note that they also include tropospheric chemistry.

P3L13: remove double "the"

removed

P3L20: "in all cases there is a peak between 2 and 5 years (not shown)." In agreement with observations?

Yes, in agreement with observations. Now noted.

P3L28: Please explain how do you get 42 model samples?

There are 3 periods and 14 model integrations. Now clarified

Table 1: Add reanalysis to the table caption

Changed to "Data Products used"

P4L10: What does it mean: "limited data was either missing, corrupted, or non-physical"?

It turns out that the issue with data quality was only applicable to CHASER, but this model has been discarded because of poor ENSO performance. We have removed this statement, and do not perform any data filling as part of this study (other than what is included in the SWOOSH product).

P7L7: When testing statistical hypotheses there could be only two outcomes, either the test is passed or not. There is no "nearly statistically significant" results. Based on your Figure 6 I conclude that the null hypothesis that the observed correlation between ASO and ENSO is accidental cannot be ruled out at the 5% rejection level.

Yes, the reviewer is correct. We have removed this statement.

P7L29: What do you mean by "the same model" here?

Clarified to "we compare adjacent 41 year sub-samples for a given model"

Figure 4: This figure makes me crazy. Correlation of correlation coefficients?!! What am I suppose to learn, for example, from the fact that there is a correlation of 0.54 between r(PS_March, ASO) and r(ZMArch_ASO)? I don't think the discussion in the text makes my task any easier. My specific concern is Figure 4d where the spread of the crosses is visually inconsistent with the reported correlation of 0.6 (for example the correlation of just 0.54 in the nearby Fig. 4c visually appears tighter). Can the authors double-check this number?

We have removed some of the extraneous information from figure 4, including the correlation of correlation information. Our discussion of this figure was overly concise in the initial submission, and we now are more thorough in describing why these correlations of correlations are meaningful.

Note that the specific correlation values have changed in the revised manuscript due to subtle changes in the code (e.g. the sub-sample periods differ slightly among others).

---

## Author Response (AR2)

Reviewer #1

The authors addressed some of the criticism and overall the manuscript has somewhat improved since the previous version. I am still not convinced that the manuscript represents a significant advance in science however I do appreciate author's efforts to disentangle statistical links between Arctic stratospheric ozone and surface climate. Below I add some comments; hopefully the authors find them useful for further improvement of the manuscript.

First, contrary to what the authors assert, I don't believe the connection between Arctic ozone and surface climate is "settled science"; I simply don't think the authors approach this problem from the right perspective. In Calvo et al they did demonstrate that, as ozone variability increased from low Ozone Depleting Substances (ODS) period to high ODS period, this was followed by increased stratospheric cooling and stronger changes in surface climate. While their study may have caveats, for example because they only use one model, they could however with reasonable confidence attribute their surface climate response during the second period to increased stratospheric variability caused by increased ODS and low ozone. I don't think they assumed a direct ozone-surface link because they carefully documented responses of zonal mean temperatures and winds consistent with the dynamical stratosphere-troposphere coupling, although they might have discussed the mechanism in a more clear way.

**We have now revised our discussion of Calvo et al in the introduction to specifically note that the downward impacts are strongest when low ozone values are more present, which is associated with stronger dynamical variability of the vortex.**

I also don't see how the authors of the present manuscript addressed the statement from Ozone assessment, which they cite. It is well-established that extreme low ozone values, observed during some winters in the Arctic, are a result of chemical depletion, rather that stratospheric dynamics alone, see for example Manney et al. 2011. Thus, based on Calvo et al results, surface anomalies that follow low ozone episodes may be attributable to chemical ozone depletion. While the ozone assessment statement does not mention any mechanism, it is certainly consistent with stratosphere-troposphere dynamical coupling similar to what is apparently operating in the Antarctic. Thus I don't see how the authors "explored the robustness" of the statement in the ozone assessment.

**If the connection between Arctic ozone and the surface is "robust", then a variety of different methodological choices should all lead to the same conclusion, namely that a downward impact is present. If the existence of a downward effect is sensitive to the specific approach one takes, then that indicates a lack of robustness.**

In summary, while I find that the part of the study concerning ASO-ENSO link has improved and is worth reporting, I still don't find that the part concerning links between ASO and polar surface climate is equally exciting. Below are my suggestions:

1. The authors find strikingly large differences in ozone-climate coupling even between simulations by the same model. In Calvo et al they find similarly large differences in coupling between two periods when ozone was / was not depleted. During ozone depletion period the ozone variability was twice as large, owing to extreme low ozone events. This was linked to larger signal in stratospheric temperatures and thus in surface climate. I suggest that the authors test whether the spread shown in Figures 4 and 6 can similarly be attributed to different interannual variability between these periods. Specifically, do you see larger interannual variability in ozone (measured e.g. by interannual standard deviation) during years when ozone-climate coupling is larger? One could expect stronger coupling when there is larger variability, due to increased signal to noise ratio.

**The RefC2 simulations aren't available for all models from 1955, and therefore we aren't able to apply the same analysis of Calvo et al but to the CCMI models. However we are able to compare the 41 year periods discussed in the text, and also the period 1985-2005 (as in Calvo et al) to the period 2072-2092 when ozone concentrations in the models are back to their values in the 1960s. This is documented in the figure below, which shows Arctic stratospheric ozone (latitude and height range as defined in the text) for each model (thin line) and the ensemble mean (thick black line). This is the time evolution of ASO before we perform the linear regression to focus on interannual timescales.**

[Figure]

**Figure R1: timeseries of ASO pressure- and area-weighted average from 50hPa to 150hPa and 60N to 81.25N in each of the 14 integrations (thin lines) and in the ensemble mean (thick black line)**

Below is a histogram of the standard deviation of March ASOpole on interannual timescales for the period 1985-2005 and 2072-2092. The top row is after regressing out greenhouse gases and ODS changes, and the bottom is when we leave in this decadal variability. Each panel also includes the standard deviation averaged across the 14 models. The standard deviation of ASOpole is actually _larger_ in 2072-2092 than in the period with elevated ODS concentrations. The interannual connection between Zpole and SLPpole is larger in the later period as well, and hence this is consistent with the reviewer's intuition. However, this stronger connection cannot be due to heterogeneous chemical ozone depletion, as there is certainly less heterogeneous chemical ozone depletion in the period 2072-2092 than in the period 1985-2005.  Results are similar if we use the 41-year periods as in the main body of the paper: the standard deviation averaged across the 14 models is 0.132 in the period 1970-2010 and 0.16 in the period 2052-2092. Note however that the mean standard deviation is somewhat less in the intervening period (0.128 for the period 2011-2051), even though the interannual relationship between Zpole and SLP was stronger in this period than in the period 1970-2010, and so there is no clear relationship between enhanced variance ASOpole and enhanced strat-trop coupling in the CCMI models. This is somewhat contrary to Calvo et al.

Calvo et al used one model – WACCM – and 3 of the 14 integrations we use here are from WACCM (though the model version is not necessarily the same). The red bars in the histogram below are for these 3 integrations of WACCM, and in the WACCM simulations as well the variance of ASOpole is larger in the period 2072-2092 than in the period 1985-2005. We do note that the increase in variance in WACCM is somewhat less than in most of the CCMI models, and hence it is possible that the difference between our analysis here and that of Calvo et al is due to the fact that Calvo et al focused on WACCM.

[Figure]

Figure R2: histogram of the standard deviation of ASO pressure- and area-weighted average from 50hPa to 150hPa and 60N to 81.25N across the 14

**integrations (top) after regressing out ODS and greenhouse gases; and (bottom) without such a regression (as in figure R1). The left column is for the 1985-2005 period with robust chemical ozone depletion, and the middle column is for the 2072-2092 period after ODS concentrations are small. The right column is the difference between the middle and left column. The average of the 14 standard deviations is indicated. Red bars indicate WACCM integrations.**

1. Further, can the authors attribute stronger coupling (and possibly larger variability) to different ozone levels and thus possibly to chemistry? The authors showed how the coupling changes between different periods (1970-2010, 2011-2051, 2052-2092) but it is difficult to understand their result without knowing what ozone values were during these runs. Is there, for example, any indication of chemical ozone depletion in those runs? One could test correlation coefficients against minimum ozone values achieved in those runs.

   **A timeseries of ozone values is shown in Figure R1, though note that for all results in the paper itself we perform a MLR analysis to statistically remove variability linearly associated with ODS and greenhouse gases. There is indeed an ozone minimum near 2000 when we do not apply the MLR.**

   **The reviewer hypothesized that models with more negative excursion of ASOpole (perhaps associated chemical ozone depletion) would be associated with enhanced coupling of ASOpole with PSpole. We now assess this hypothesis in the CCMI models in figure R3 below. Specifically, we compute minimum value in ASOpole in March in each integration relative to its mean value over each 41 year chunk for each model. This minimum value is shown in the x-axis. The y-axis shows the correlation of PSpole with ASO, with the top row the correlation of PSpole in March with ASO and the bottom row the correlation of PSpole in April with ASO. Each star is for one of the 14 integrations. If the reviewer's intuition is correct, we would expect a negative correlation, and in 3 of the 6 there is indeed a negative correlation (in the other 3 the correlation is essentially zero). However none of these negative correlations are statistically significant at even the 90% level. Hence while there is some indication that the reviewer's hypothesis is supported by the CCMI REFC2 runs, it is premature to include any of**

**this analysis in the paper itself as results aren't statistically significant.**

[Figure]

**Figure R3: Comparison of the (x-axis) minimum value of ASOpole for each model and each 41 year chunk as compared to the mean value for that period to (y-axis) the correlation of ASOpole in March with PSpole in March (top) and April (bottom).**

I also wonder what could be the cause for seemingly stronger coupling during some future runs (2052-2092). One could imagine that there are still episodes of chemical ozone depletions that, combined with increased Brewer-Dobson circulation in future, would lead to larger variability and stronger coupling. I guess negative answer to these questions would indeed indicate that the variability in coupling is due to internal variability, as the authors assert, but the tests proposed above could still give more insights.

**This is a very good question, and we don't have a good answer. The variance of ASOpole is indeed larger in the 2052-2092 period, and so is the variance in Zpole and also in Tpole (see Figure R4, with the figures constructed similar to Figure R2). However ODS concentrations are less in the period 2052-2092, and so this increased variance does not appear to be related to heterogeneous chemical ozone depletion. Note that the increases in variance for Zpole, Tpole, and ASO are all between 15% and 20%. Ultimately, we leave a thorough explanation for this finding for future work.**

[Figure]

**Figure R4: histogram of the standard deviation of polar cap geopotential height and temperature at 100hPa area-weighted average from 70N to the pole across the 14 integrations after regressing out ODS and greenhouse gases. The left column is for the 1970-2010 period with robust chemical ozone depletion, the middle column is for the 2011-2052 period, and the right column is after ODS concentrations are small. The average of the 14 standard deviations is indicated.**

I appreciate large amount of work that authors put into the study but I nevertheless don't find that the manuscript is complete without this additional (and rather straightforward) analysis.

**We have made three changes to the manuscript:**

1. **The discussion of Calvo et al 2015 in the introduction has been improved: " In contrast, the coupled chemistry-climate simulations of Calvo et al. (2015) include a robust stratospheric-tropospheric response in low versus high ozone years if ozone depleting substance concentrations (ODS) follow those from 1985-2005: a positive phase of the North Atlantic Oscillation, a poleward shift of the North Atlantic tropospheric jet, and corresponding regional surface temperature anomalies. Downward coupling is not evident in the period 1955-1975 when ODS concentrations were lower, and they argue that the enhanced ODS concentrations lead to enhanced dynamical variability of the vortex and thus to stronger downward coupling. The fully-coupled**

approach of Calvo et al. (2015) allows consistency between the evolving ozone distributions and dynamical conditions among other differences in the model configuration, which may explain the differences between their conclusions and those of studies prescribing ozone concentrations"

2. We have revised the end of section 3 to better emphasize that the connection between vortex variability and surface climate is stronger in a nearly ODS-free future than when ODS concentrations neared their peak.

3. We have also added to the discussion section: " Second, Calvo et al. (2015) argue that chemical ozone depletion leads to a more variable vortex and specifically to more extremely strong vortex events, which in turn leads to more robust stratosphere-troposphere coupling. However we find that  variability of the vortex is actually larger in the future when ODS concentrations are lower and heterogeneous chemical ozone depletion should be less frequent. Specifically, the standard deviation averaged across all models for the period 1970-2010 is lower than for the period 2052-2092 for ASO, polar cap geopotential height, and polar cap height by 15% to 20%, and this increase in variability is driven both by more frequent positive and negative extremes. A thorough investigation of what drives this enhanced future stratospheric vortex variability in the CCMI models is beyond the scope of this work, though we note that Ayarzagüena et al. (2018) find little future change in sudden warming frequency in these models."

We have elected not to include more of the new analysis in this response in the text, as results were either difficult to interpret or not statistically significant.  We have added one new figure to the text that better emphasizes that dynamical coupling is far stronger than any coupling between ASO and the surface however (see below). Because this dynamical coupling is so strong, trying to find evidence for the role of chemical ozone depletion separately from that of transport is rather difficult, though there is certainly room for future work.

[Figure]

**Figure 5.** A scatter plot of (y-axis) polar cap surface pressure with (x-axis; top) March polar cap geopotential height and (x-axis; bottom) Arctic stratospheric ozone. April polar cap surface pressure is used for the top row, and March polar cap surface pressure is used for the bottom row, corresponding to the month with strongest coupling. Each year of each CCMI integration is marked with a diamond. All correlations are statistically significant at the 95% level as given by a two-tailed student-t test.

Reviewer #2

Suggestions for revision or reasons for rejection (will be published if the paper is accepted for final publication)

Overall, my major concerns have been addressed and I recommend publication. I do have a few very minor comments/edits that the authors might address.

Page 5, Line 24: I don't understand which part of this is named after the developers, or why it matters?

**Fixed. The reason it matters is that PC could also stand for partial correlation as in step 2, but in this case it does not.**

Page 5, Line 31: put a space between Linear and Mediation

**Fixed.**

Figure 2: note that some of the negative contours are not filled in and it would visually look better to fully saturate those colors.

**Fixed.**

Page 6, Line 14 and lines 31-32 are repetitive

**We elected to leave the text as is.**

Page 8, line 13-14: doesn't this speak more to non-stationarity than non-linearities?

**We removed "nonlinear". The key point is that the connection between vortex variability and surface climate is stronger in the absence of ODS. We have revised the text accordingly**.

Page 8, 17: not clear which relationship you are talking about here- ASO on surface climate?

**Fixed.**

[revised manuscript text omitted]